# Synthetic Text Generation for Training Large Language Models via Gradient Matching

**Dang Nguyen** [* 1 2]   **Zeman Li** [* 2 3]   **Mohammadhossein Bateni** [2]   **Vahab Mirrokni** [2]   **Meisam Razaviyayn** [2 3]
**Baharan Mirzasoleiman** [1 2]

## Abstract

Synthetic data has the potential to improve the performance, training efficiency, and privacy of real training examples. Nevertheless, existing approaches for synthetic text generation are mostly heuristics and cannot generate human-readable text without compromising the privacy of real data, or provide performance guarantees for training Large Language Models (LLMs). In this work, we propose the first theoretically rigorous approach for generating synthetic human-readable text that provides convergence, performance, and privacy guarantees for fine-tuning LLMs on a target task. To do so, we leverage Alternating Direction Method of Multipliers (ADMM) that iteratively optimizes the embeddings of synthetic examples to match the noisy gradient of the target training or validation data, and maps them to a sequence of text tokens with low perplexity. In doing so, the generated synthetic text guarantees convergence of the model to a close neighborhood of the solution obtained by fine-tuning on real data and preserves their privacy. Experiments on various classification tasks confirm the effectiveness of our proposed approach. Our code is available at https://github.com/BigML-CS-UCLA/GRADMM.

## 1. Introduction

High-quality data is crucial for training Large Language Models (LLMs) to superior performance (Yang et al., 2024; Li et al., 2023b). However, collecting and curating high-quality data is often very expensive and hard to obtain in many domains. In addition, as LLMs can memorize their

---
[*]Equal contribution  [1]Department of Computer Science, University of California, Los Angeles [2]Google Research [3]Department of Industrial & Systems Engineering, University of Southern California. Correspondence to: Dang Nguyen <dangnth@cs.ucla.edu>.

*Proceedings of the 42nd International Conference on Machine Learning*, Vancouver, Canada. PMLR 267, 2025. Copyright 2025 by the author(s).

training data (Hartmann et al., 2023), ensuring the privacy of training examples hinders training the model directly on the training data. Thus, generating small subsets of synthetic data that can train an LLM to superior performance on the target task becomes handy. To do so, synthetic text should be generated in a way that ensures similar dynamics to that of training on the real data. However, text is discrete in nature and optimization in the discrete space is very challenging.

Existing approaches for synthetic text generation mostly rely on advanced LLMs such as GPT-4 to generate synthetic text for the target categories (Ye et al., 2022; Meng et al., 2022; Li et al., 2023b; Gupta et al., 2023; Tao et al., 2024; Wu et al., 2024; Dekoninck et al., 2024; Yu et al., 2024). LLM-generated text either suffers from lack of diversity and faithfulness to real data (Ye et al., 2022; Meng et al., 2022; Li et al., 2023b), or requires meticulous prompt engineering and highly complex pipelines, such as multi-agent frameworks, iterative sampling, and processing mechanisms (Gupta et al., 2023; Dekoninck et al., 2024; Wu et al., 2024). The complexity of the pipelines, efforts for manual prompt engineering, and the cost of querying advanced models limits the applicability of such approaches. A few recent studies explored the use of VAEs and diffusion for controllable text generation (Li et al., 2022; Gong et al., 2022; Zhou et al., 2024). But, training diffusion models is computationally heavy and difficult in practice. Importantly, none of the above approaches guarantees performance of LLMs trained on the synthetic text or preserve privacy of real data.

The above limitations raise a key question: *Can we generate a small subset of synthetic text that can train an LLM with similar dynamics to that of real data?* For vision models, Dataset Distillation (DD) addresses the above question by generating a small number of synthetic images that minimize the training loss (Wang et al., 2018; Loo et al., 2022; Nguyen et al., 2020), match the training gradient (Zhao et al., 2020; Zhao & Bilen, 2021) or model's weight trajectory during training (Cazenavette et al., 2022; Wang et al., 2022). For images, gradient-based methods can easily operate in the pixel-wise continuous space. However, for LLMs, the discrete nature of text and the very large number of LLM's parameters make DD much more challenging. The few ex-

isting approaches generate synthetic embeddings that minimize the training loss (Sucholutsky & Schonlau, 2021; Li & Li, 2021; Sahni & Patel, 2023; Maekawa et al., 2023), or by training a generator model to match the gradient of an LLM trained on target data (Maekawa et al., 2024). However, the synthetic embeddings are not readable and cannot be transferred to train other LLMs, and synthetic data generated by matching dynamics of a LLM trained on target data may include real training examples and is not privacy preserving.

In this work, we propose the first theoretically-rigorous method to generate readable synthetic text that guarantees similar dynamics to that of fine-tuning on real data. First, we formulate a discrete optimization problem to find text embeddings that have a similar gradient to that of real data, under the constraint that the optimized embeddings should correspond to tokens in the vocabulary. Moreover, to ensure readability, we add another constraint that requires the sequence to have a low perplexity. Then, we solve this discrete optimization problem using Alternating Direction Method of Multipliers (ADMM) that iteratively optimizes the embeddings of synthetic data to match the average gradient of the real data, and maps them to a sequence of text tokens with low perplexity. To guarantee Differential Privacy (DP), we clip real data gradients and add controlled noise to their average before matching it. We prove that the synthetic text generated by our method guarantees convergence to a close neighborhood of the solution obtained by fine-tuning the model on real data.

We conduct extensive experiments to evaluate the effectiveness of our approach, namely GRADMM, for generating synthetic data using Phi model for multiple classification tasks. First, we consider the case where only a small number of validation examples are available and we apply GRADMM to generate a larger fine-tuning data. We show that with only 5 to 50 examples, GRADMM can successfully generate 100 synthetic data that outperform training on the real examples by up to 31.5%. Next, we apply GRADMM to generate a small synthetic data based on an existing larger fine-tuning data. We show that the synthetic data generated by GRADMM outperforms zero-shot and few-shot generation by LLMs as well as real examples selected by coreset selection methods by up to 13.1%, while ensuring the privacy of the training data. We also confirm the transferability of GRADMM's generated text via Phi for fine-tuning other LLMs, including Llama-3.2-1B and OPT-1.3B.

## 2. Related Work

### 2.1. Dataset Distillation (DD)

DD aims to generate a small synthetic subset of examples that can achieve a similar generalization performance to that of training on the full real dataset.

**DD for Images.** DD is originally proposed for images. Wang et al. (2018) initially proposed a meta-learning approach which synthesizes data by iteratively training a model to convergence on the synthetic examples, and optimizing the synthetic data such that the trained model generalizes well on the real training data. Subsequent studies tried to make this process more efficient by using kernel methods to approximate training the model on synthetic data in a closed form (Loo et al., 2022; Nguyen et al., 2020). More recent works generate synthetic data by matching the gradient (Zhao et al., 2020; Zhao & Bilen, 2021; Kim et al., 2022) or wright trajectory (Cazenavette et al., 2022; Wang et al., 2022) of the model trained on real data, or by matching the data distribution (Zhao & Bilen, 2023).

**DD for Text.** There have been recent efforts in applying DD to text. For text datasets, existing methods (Sucholutsky & Schonlau, 2021; Li & Li, 2021; Sahni & Patel, 2023) apply the original meta-learning based method of (Wang et al., 2018), or minimize the KL-divergence between the self-attention probabilities of the model and the distilled attention labels across all layers and heads, for the first token Maekawa et al. (2023). As generating text in the discrete space is difficult, the synthetic data is generated as continuous input word embeddings instead of discrete text. Such embeddings cannot be used for training other models that have different word embedding weights, and are unreadable to humans, making them difficult to interpret and analyze. Sucholutsky & Schonlau (2021); Sahni & Patel (2023) transformed their distilled synthetic samples to text by finding words with the nearest neighbor embeddings. However, this results in unrelated words that are not meaningful.

To generate readable text, Maekawa et al. (2024) first trains a proxy language model from scratch to generate synthetic training data for different classes. Then, it fine-tunes a generator model to generate synthetic data by minimizing the gradient matching loss between generated and training data. Training the proxy model is a bottleneck in scaling the method. Besides, as the distilled synthetic data may include real samples from the original dataset, this method cannot ensure privacy.

Notably, none of the existing DD methods scale beyond BERT (Devlin, 2018) to LLMs with billions of parameters. In this work, we propose the first DD method that can generate privacy-preserving human-readable text, by matching gradients of LLMs with billions of parameters.

### 2.2. Synthetic Text Generation using Generative Models

**LLMs.** A large body of recent work used LLMs to generate synthetic text data in the zero-shot or few shot setting (Meng et al., 2022; Li et al., 2023b). In the zero-shot setting, the LLM is directly prompted to generate text for categories of interests. In the few-shot setting, a few

real-world data instances are provided as examples to guide the LLM in generating the synthetic data. In our work, we use the zero-shot and few-shot approaches as our baselines. LLM-generated text is often very repetitive and lacks diversity (Holtzman et al., 2019; Keskar et al., 2019). Besides, it does not capture the distribution of the target task and may contain incorrect or hallucinated examples (Ye et al., 2022; Meng et al., 2022; Gupta et al., 2023; Li et al., 2023b; Wu et al., 2023). To address these issues, recent methods rely on extensive prompt engineering to inject semantic diversity for each target category (Gupta et al., 2023) and design highly complex pipelines, such as model arithmetic which composes and biases multiple LLMs (Dekoninck et al., 2024), multi-step meticulous prompt engineering to inject domain knowledge, iterative sampling, and self-correction to rectify inaccurately labeled instances (Gupta et al., 2023), and retrieval-augmented generation techniques (Wu et al., 2023). Such pipelines require a large number of queries to advanced LLMs such as GPT-4 (OpenAI, 2023) and Claude3-Opus (Anthropic, 2023). This incurs a large financial cost and makes such approaches difficult to apply in practice. While synthetic data generated by LLMs are human-readable, LLMs may memorize and generate their training data (Hartmann et al., 2023). Hence, the synthetic data generated by LLMs do not preserve the privacy of their training data. Besides, it does not provide any theoretical guarantee for the performance of LLMs trained on them.

**VAE and Diffusion.** A few recent studies explored the use of VAEs and diffusion for controllable text generation (Li et al., 2022; Gong et al., 2022; Zhou et al., 2024). Such approaches train a diffusion model from scratch and parameterize structural and semantic controls by different classifiers and update the latent variables to satisfy the controls (Li & Li, 2021), or to add noise and denoise embeddings of real data (Zhao et al., 2020). This process is computationally very heavy and difficult in practice. Similar to LLMs, synthetic data generated by VAEs and diffusion models do not provide guarantee for the performance of the trained model.

## 3. Problem Formulation

Here, we formalize the problem of generating small human-readable synthetic text that can fine-tune an LLM with similar dynamics to that of real data. We also discuss two common use cases where such synthetic data is useful.

**Setting.** Consider a pretrained LLM with parameters $\boldsymbol{\theta}$ and vocabulary $V = \{v_1, \cdots, v_{|V|}\}$ containing all the words it has been trained to recognize and use. Consider a supervised fine-tuning dataset $\mathcal{D}_T = \{s^i\}$, where each example $s^i = (\boldsymbol{p}^i, \boldsymbol{r}^i)$ is a pair of prompt $\boldsymbol{p}^i$ and response $\boldsymbol{r}^i$ containing words in the vocabulary. The negative log likelihood loss is defined as $\ell(s^i, \boldsymbol{\theta}) = -\log(\boldsymbol{r}^i|\boldsymbol{p}^i)$. The fine-tuning objective is thus to minimize the negative log likelihood loss

over the whole dataset $\mathcal{D}$ as $\ell(\mathcal{D}, \boldsymbol{\theta}) = \frac{1}{|\mathcal{D}|}\sum_{i=1}^{|\mathcal{D}|}\ell(s^i, \boldsymbol{\theta})$.

**Problem formulation.** Given a subsets of real examples from the fine-tuning data $\mathcal{D}_{\text{real}} \subset \mathcal{D}_T$, our goal is to generate synthetic data $\mathcal{D}_{\text{syn}} = \{q^i\}_{i=1}^k, q^i \notin \mathcal{D}_T \ \forall i$, containing $r$ synthetic examples that do not belong to $\mathcal{D}_T$, such that fine-tuning the model on $\mathcal{D}_{\text{syn}}$ minimizes the loss on $\mathcal{D}_{\text{real}}$. Formally,

$$\underset{\mathcal{D}_{\text{syn}}, |\mathcal{D}_{\text{syn}}|\leq r}{\arg\min} \ \ell(\mathcal{D}_{\text{real}}, \boldsymbol{\theta}^*), \quad s.t. \quad \boldsymbol{\theta}^* \in \underset{\boldsymbol{\theta}}{\arg\min} \ \ell(\mathcal{D}_{\text{syn}}, \boldsymbol{\theta}). \tag{1}$$

**Readability constraint.** Importantly, we want the synthetic data to be human-readable. That is we want every synthetic example to be a sequence of words in the vocabulary. Besides, to ensure that the sequence is meaningful, we require that the synthetic data has low perplexity. Thus, we wish to solve the following constrained optimization problem:

$$\underset{\substack{\mathcal{D}_{\text{syn}}, |\mathcal{D}_{\text{syn}}|\leq k, \\ s\in\Gamma, \text{ppl}(s)\leq\epsilon \\ \forall s\in\mathcal{D}_{\text{syn}}}}{\arg\min} \ \ell(\mathcal{D}_{\text{real}}, \boldsymbol{\theta}^*), \quad s.t. \quad \boldsymbol{\theta}^* \in \underset{\boldsymbol{\theta}}{\arg\min} \ \ell(\mathcal{D}_{\text{syn}}, \boldsymbol{\theta}), \tag{2}$$

where $\Gamma = \{s = (\boldsymbol{p}, \boldsymbol{r})|p_j, r_j \in V\}$ is the set of all prompts and responses that consist of words in vocabulary $V$.

**Use cases for synthetic data generations.** The above formulation is applicable to two settings: (1) Data is scarce for the target task, and we want to generate a larger synthetic fine-tuning data based on a small number of examples from the target task. (2) A relatively large supervised fine-tuning data is available, and we wish to generate a smaller synthetic data to replace the real data to preserve the privacy of training examples or to improve the training efficiency.

## 4. Method

Next, we discuss our proposed method for generating readable synthetic text for fine-tuning LLMs on a target task.

### 4.1. Text Generation via Gradient Matching

An effective way to solve Eq 2 is via gradient matching. Specifically, we generate a synthetic data $\mathcal{D}_{\text{syn}}$ that has a similar gradient to that of the real dataset:

$$\underset{\substack{\mathcal{D}_{\text{syn}}, |\mathcal{D}_{\text{syn}}|\leq r, \\ s\in\Gamma, \text{ppl}(s)\leq\epsilon \\ \forall s\in\mathcal{D}_{\text{syn}}}}{\arg\min} \ D(\nabla_{\boldsymbol{\theta}}\ell(\mathcal{D}_{\text{syn}}, \boldsymbol{\theta}), \nabla_{\boldsymbol{\theta}}\ell(\mathcal{D}_{\text{real}}, \boldsymbol{\theta})). \tag{3}$$

where $D(\cdot, \cdot)$ is a distance between two gradients. Following (Deng et al., 2021; Geiping et al., 2020), we use $1 - cos(., .)$ as our distance metric, where $cos$ is the cosine similarity. If such a synthetic data can be generated, training on it with gradient methods directly minimizes the loss on real data.

Fine-tuning is often short and changes the model to a smaller extent than pre-training. Fine-tuning for longer results in forgetting the pretrained information and harms the performance (Gekhman et al., 2024). Since fine-tuning loss is often smooth and has a bounded curvature, we solve the above problem by generating a synthetic data that matches the gradient of real data at the pretrained parameters. We prove that training on such a subset converges to a close neighborhood of the solution found by training on real data.

**Challenges of Readable Text Generation.** Solving Problem 3 is very challenging, as the set of feasible solutions is sparse, the space is discrete, and LLMs are non-linear and high-dimensional. Specifically, the constraint set is formed by the Cartesian product of many discrete sets, each restricting a word to belong to the vocabulary. Among sequences that satisfy this condition, only those that are readable—measured by a low perplexity value—are valid. Thus, solving Problem 3 is NP-hard as it requires going through all the possible sequences of words in the vocabulary and finding readable sequences that best match the real data gradient. The number of such sequences is exponential in the size of the vocabulary. This makes it computationally infeasible to find the optimum solution. Finally, calculating the similarities in the gradient space of LLMs with billions of parameters is computationally very expensive.

### 4.2. Alternating Between Text and Embedding Spaces

To solve the above discretely constrained non-convex optimization problem, we first transfer it to the continuous embedding space, where one can optimize the embeddings of synthetic data to match the target gradient, under the constraint that the optimized embeddings belong to the set of all token (words, subwords, or characters) embeddings in the vocabulary. If such embeddings can be found, they can be directly mapped to a sequence of words in the vocabulary.

Formally, let $\boldsymbol{x} \in \mathbb{R}^{n \times d}$ be the embedding matrix of a synthetic sample $s$ with $n$ tokens, where row $x_j \in \mathbb{R}^d$ is the $j^{th}$ token embedding. By stacking the embedding matrices of all synthetic samples in $\mathcal{D}_{\text{syn}}$, we obtain an embedding tensor $\boldsymbol{X} \in \mathbb{R}^{|\mathcal{D}_{\text{syn}}| \times n \times d}$. With an abuse of notation, we denote $\ell(\boldsymbol{x}, \boldsymbol{\theta}) = \ell(s, \boldsymbol{\theta})$ and $\ell(\boldsymbol{X}, \boldsymbol{\theta}) = \frac{1}{|\mathcal{D}_{\text{syn}}|} \sum_{i=1}^{|\mathcal{D}_{\text{syn}}|} \ell(\boldsymbol{x}^i, \boldsymbol{\theta})$. We rewrite Problem 3 as:

$$\underset{\substack{\mathcal{D}_{\text{syn}}, |\mathcal{D}_{\text{syn}}| \leq r, \\ x_j \in \mathcal{E}, \text{ppl}(\boldsymbol{x}) \leq \epsilon \\ \forall \boldsymbol{x} \in \mathcal{D}_{\text{syn}}}}{\arg \min} f(\boldsymbol{X}) \ \text{ s.t. } \ f(\boldsymbol{X}) = D(\nabla_{\boldsymbol{\theta}} \ell(\boldsymbol{X}, \boldsymbol{\theta}), \nabla_{\boldsymbol{\theta}} \ell(\mathcal{D}_{\text{real}}, \boldsymbol{\theta})),$$

(4)

where $\mathcal{E} = \{e_1, e_2, \ldots, e_{|V|}\}$ denote the vocabulary embedding, i.e. the set of all token embeddings in the vocabulary $V$ of model $\boldsymbol{\theta}$ where $e_i \in \mathbb{R}^d$ and $d$ is the embedding dimension.

To solve the above constrained optimization problem we apply the Alternating Direction Method of Multipliers

(ADMM) (Glowinski & Marroco, 1975; Gabay & Mercier, 1976). By forming the augmented Lagrangian function, ADMM decomposes the original problem into subproblems that can be solved separately and iteratively. While ADMM was originally introduced for convex optimization under linear constraints, more recently it has been successfully applied to solving mixed integer non-linear programs (Leng et al., 2018; Lin et al., 2019), with convergence guarantees (Huang et al., 2021).

**Constrained Gradient Matching in the Embedding Space.** To apply ADMM to our discretely constrained non-convex problem 4, we convert it to a non-convex optimization with convex linear constraints. To do so, we introduce an auxiliary variable $\boldsymbol{Z}$ and rewrite our objective with an extra equality constraint so that the embeddings are constrained to be from the vocabulary, but not subject to that restriction:

$$\min_{\boldsymbol{X}} f(\boldsymbol{X}) + \mathcal{I}_{\mathcal{E}}(\boldsymbol{Z}), \quad s.t. \quad \boldsymbol{X} = \boldsymbol{Z}. \tag{5}$$

The indicator function $\mathcal{I}_{\mathcal{E}}(\boldsymbol{Z})$ is defined as $\mathcal{I}_{\mathcal{E}}(\boldsymbol{Z}) = 0$ if $z_j \in \mathcal{E} \ \forall j$ (i.e., if the embedding of each synthetic example can be mapped to a sequence of words in the vocabulary), and $\mathcal{I}_{\mathcal{E}}(\boldsymbol{Z}) = +\infty$ otherwise. The augmented Lagrange of Eq. 5 for parameter $\rho > 0$, can be formulated as:

$$\mathcal{L}_{\text{aug}}(\boldsymbol{X}, \boldsymbol{Z}, \boldsymbol{\Lambda}) = f(\boldsymbol{X}) + \mathcal{I}_{\mathcal{E}}(\boldsymbol{Z}) + \langle \boldsymbol{\Lambda}, \boldsymbol{X} - \boldsymbol{Z} \rangle \\ + \frac{\rho}{2} \|\boldsymbol{X} - \boldsymbol{Z}\|^2, \quad (6)$$

where $\boldsymbol{\Lambda} \in \mathbb{R}^{|\mathcal{D}_{\text{syn}}| \times n \times d}$ denotes the Lagrangian multipliers. With simple algebraic manipulations, Eq.6 can be written as:

$$\mathcal{L}_{\text{aug}}(\boldsymbol{X}, \boldsymbol{Z}, \boldsymbol{\Lambda}) = f(\boldsymbol{X}) + \mathcal{I}_{\mathcal{E}}(\boldsymbol{Z}) + \frac{\rho}{2} \|\boldsymbol{X} - \boldsymbol{Z} - \rho^{-1} \boldsymbol{\Lambda}\|^2. \tag{7}$$

ADMM solves the above problem by minimizing primal variables $\boldsymbol{X}, \boldsymbol{Z}$ and maximizing dual variable $\boldsymbol{\Lambda}$ at each iteration $t$, using the following update rules:

Primal update: $\boldsymbol{X}^{t+1} = \underset{\boldsymbol{X}}{\arg \min} \ \mathcal{L}_{\text{aug}}(\boldsymbol{X}, \boldsymbol{Z}^t, \boldsymbol{\Lambda}^t),$ (8)

$$\boldsymbol{Z}^{t+1} = \underset{\boldsymbol{Z}}{\arg \min} \ \mathcal{L}_{\text{aug}}(\boldsymbol{X}^{t+1}, \boldsymbol{Z}, \boldsymbol{\Lambda}^t), \quad (9)$$

Dual update: $\boldsymbol{\Lambda}^{t+1} = \boldsymbol{\Lambda}^t + \rho(\boldsymbol{X}^{t+1} - \boldsymbol{Z}^{t+1}),$ (10)

which are respectively the proximal step, projection step, and dual update. The proximal step optimizes the embeddings to match the target gradient, and the projection step maps the embeddings to words in the vocabulary. Eq. 8 requires solving an unconstrained optimization problem. When $\rho$ is large, the function is strongly convex in $\boldsymbol{X}$. In practice, stochastic gradient descent algorithms such as Adam (Kingma, 2014) can obtain an approximate solution, which is sufficient for the convergence of ADMM (Huang et al., 2021). Next, we discuss the projection step.

**Algorithm 1** GRADient matching w. ADMM (GRADMM)

---

1: **Input:** Constant $\rho > 0$, ADMM steps $T$, proj param $k$, DP param $\varepsilon, \delta$
2: **Step 1: Initialization**
3: Random sample $\boldsymbol{X} \in \Gamma$
4: Initialize $\boldsymbol{X}^0 = \arg\min_{\boldsymbol{X}} f(\boldsymbol{X}, \varepsilon, \delta)$
5: Initialize $\boldsymbol{Z}^0 = \boldsymbol{X}^0$ and $\boldsymbol{\Lambda}^0 \in \mathbb{R}^{|\mathcal{D}_{\text{syn}}| \times n \times d}$.
6: **Step 2: ADMM**
7: **for** $t = 0, 1, \ldots, T-1$ **do**
8:    Update $\boldsymbol{X}$: $\boldsymbol{X}^{t+1} = \arg\min_{\boldsymbol{X}} \mathcal{L}(\boldsymbol{X}, \boldsymbol{Z}^t, \boldsymbol{\Lambda}^t, \varepsilon, \delta)$
9:    Update $\boldsymbol{Z}$: $\boldsymbol{Z}^{t+1} = \mathcal{P}_{\mathcal{E}_{\text{top-k}}}(\boldsymbol{X}^{t+1} + \rho^{-1}\boldsymbol{\Lambda}^t)$
10:    Update $\boldsymbol{\Lambda}$: $\boldsymbol{\Lambda}^{t+1} = \boldsymbol{\Lambda}^t + \rho(\boldsymbol{X}^{t+1} - \boldsymbol{Z}^{t+1})$
11: **end for**
12: $\mathcal{S} = \mathcal{P}_{\mathcal{E}_{\text{top-k}}}(\boldsymbol{X}^T)$
13: **Step 3: Filtering**
14: Drop samples in $\mathcal{S}$ that do not belong to their category
15: Select $r$ samples in $\mathcal{S}$ with lowest gradient matching loss
16: Drop examples with highest loss from categories that have a higher average gradient matching loss
17: **Output:** Remaining synthetic texts in $\mathcal{S}$.

---

**Projecting the Embeddings into the Vocabulary Space.** Equation (9) can be written as (Huang et al., 2021):

$$\boldsymbol{Z}^{t+1} = \arg\min_{\boldsymbol{Z}} \mathcal{L}_{\text{aug}}(\boldsymbol{Z}^{t+1}, \boldsymbol{Z}, \boldsymbol{\Lambda}^t) \tag{11}$$

$$= \arg\min_{\boldsymbol{Z}} \mathcal{I}_{\mathcal{E}}(\boldsymbol{Z}) + \|\boldsymbol{Z} - \boldsymbol{X}^t - \rho^{-1}\boldsymbol{\Lambda}^t\|^2 \tag{12}$$

$$= \mathcal{P}_{\mathcal{E}}(\boldsymbol{X}^t + \rho^{-1}\boldsymbol{\Lambda}^t). \tag{13}$$

For the vocabulary embeddings $\mathcal{E}$, the projection $\mathcal{P}_{\mathcal{E}}(x_i)$ of an embedding vector $x_i \in \mathbb{R}^d$ into the vocabulary space is the embedding vector $z_i := \arg\min_{e \in \mathcal{E}} \|x_i - e\|^2$ corresponding to the token in the vocabulary that is closest to $x_i$ in Euclidean space. In practice, $z_i$ can be found by looping over the vocabulary and finding the closest token. This operation can be vectorized efficiently. For an embedding matrix $\boldsymbol{x} \in \mathbb{R}^{n \times d}$ consisting of $n$ embedding vectors, we project each embedding vector $x_i$ independently to get the matrix embedding $\mathcal{P}_{\mathcal{E}}(\boldsymbol{x}) = \begin{bmatrix} \mathcal{P}_{\mathcal{E}}(x_1) & \mathcal{P}_{\mathcal{E}}(x_2) & \cdots & \mathcal{P}_{\mathcal{E}}(x_n) \end{bmatrix}^\top$. Similarly, for the embedding tensor $\boldsymbol{X}$, the projection operation can be vectorized efficiently to find $\boldsymbol{Z}$.

**Ensuring Readability of the Projected Text.** Projecting embeddings to tokens in vocabulary independently does not yield meaningful text. To address this, we leverage the idea of top-$k$ decoding to enforce the readability of generations (Fan et al., 2018). Consider an embedding matrix $\boldsymbol{x} \in \mathbb{R}^{n \times d}$ consisting of $n$ embedding vectors. For every embedding $x_i$, we find the top $k$ most probable tokens from the vocabulary condition on the previously projected tokens. Formally, we find the top $k$ tokens that minimize $\sum_{e \in \mathcal{E}} P(x|x_{i=1:i-1})$, and denote them by $\mathcal{E}_{\text{top-k}}$. Then, we project $x_i$ into the space of the top-$k$ selected tokens by solving $z_i := \mathcal{P}_{\mathcal{E}_{\text{top-k}}}(x_i) = \arg\min_{e \in \mathcal{E}_{\text{top-k}}} \|x_i - e\|^2$.

## 4.3. Dealing with High-dimension Gradients

Calculating similarities in the very high-dimensional gradient space of LLMs with billions of parameters is computationally very expensive. Besides, such gradients contain many small and noisy dimensions which makes calculating gradient similarities inaccurate. An effective way to tackle this issue is to leverage lower-dimensional gradient estimates (Mirzasoleiman et al., 2020). Various weight initialization (Glorot & Bengio, 2010) and activation normalization methods (Ioffe, 2015) uniformize the activations across samples. Thus, the variation of the gradient norm is mostly captured by the gradient of the loss with respect to the model's last layer (Katharopoulos & Fleuret, 2018).

Based on the observation, we generate synthetic data by only matching the last-layer gradient of the model. Let $\boldsymbol{\theta}_L$ denote the last layer of model $\boldsymbol{\theta}$ with $L$ layers, the last-layer gradient distance between synthetic and real data in Eq 4 is:

$$\underset{\substack{\mathcal{D}_{\text{syn}}, |\mathcal{D}_{\text{syn}}| \leq r, \\ x_j \in \mathcal{E}, \text{ppl}(\boldsymbol{x}) \leq \epsilon \\ \forall \boldsymbol{x} \in \mathcal{D}_{\text{syn}}}}{\arg\min} \quad D(\nabla_{\boldsymbol{\theta}} \ell(\boldsymbol{X}, \boldsymbol{\theta}_L), \nabla_{\boldsymbol{\theta}} \ell(\mathcal{D}_{\text{real}}, \boldsymbol{\theta}_L)) \tag{14}$$

Matching the last layer gradient is much cheaper than the full gradient and allows generating synthetic data with superior performance, as we will confirm in our experiments.

## 4.4. Filtering the Generated Examples

While the top-$k$ projection enables generating human-readable text, it can negatively affect the performance due to the following reasons: (i) It may change the category of the synthetic example by including words that are most relevant to other categories. (ii) It may significantly increases the gradient matching loss of some synthetic examples. (iii) It may result in a much higher gradient matching loss for some categories compared to the rest. To address the above issues, we filter the low-quality synthetic examples as follows. First, we drop examples that do not belong to the correct category by running a simple few-shot evaluation (Li et al., 2023b), as detailed in Appendix B. Next, for every category we select $r$ synthetic examples with the lowest gradient matching loss. Finally, we ensure similar gradient matching loss for all categories by dropping examples with highest loss in categories with a higher average loss compared to the rest. The above steps significantly boost the performance of the synthetic data, as we will confirm in our experiments.

***Remark.*** Due to top-$k$ decoding, the filtered synthetic examples do not match the target gradient very accurately. Thus, we generate each synthetic example to match the target gradient independently, not conditioned on each other. We will confirm in our experiments in Sec. 5.3 that the synthetic data generated independently by GRADMM matches the real data gradient closely during fine-tuning.

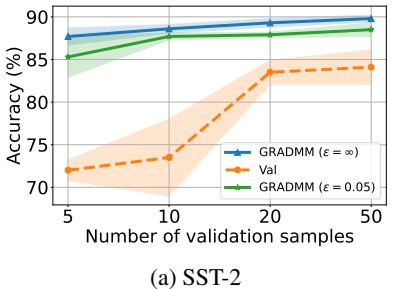
(a) SST-2

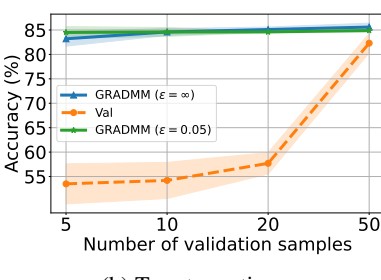
(b) Tweet emotions

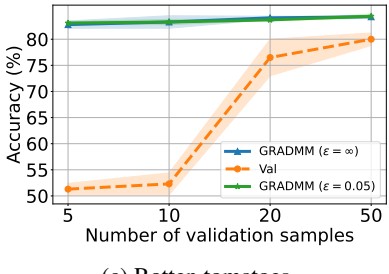
(c) Rotten tomatoes

*Figure 1.* Data-scarce regime. Generating 100 synthetic samples with GRADMM, based on 5, 10, 20, 50 examples from a target task. Synthetic data generated based on only 5 real examples outperforms the real data by 15.7%, 29.7%, and 31.5% on the three datasets.

### 4.5. Making GRADMM Differentially Private

Differential privacy (DP) (Dwork et al., 2006) is a rigorous mathematical framework that ensures no single data point can be identified or inferred from the output of a statistical or machine learning model. To make GRADMM differentially private, we inject controlled noise $\alpha$ into the clipped gradient of the real data in Equation (4). Specifically, GRADMM first computes per-sample gradients of the real data and clips their $\ell_2$-norm to a threshold of $C$. These clipped gradients are then averaged, and Gaussian noise, drawn from $\mathcal{N}(0, \sigma^2)$, is added to this average. The added noise scale $\sigma$ is defined as follows:

$$\sigma = \begin{cases} \frac{C\sqrt{2\log\frac{1.25}{\delta}}}{\varepsilon|\mathcal{D}_{\text{real}}|}, & \text{if } 0 < \varepsilon \leq 1 \text{ (Dwork et al., 2014)} \\ \frac{C(c+\sqrt{c^2+\varepsilon})}{\sqrt{2}\varepsilon|\mathcal{D}_{\text{real}}|}, & \text{if } \varepsilon > 1 \text{ (Lowy \& Razaviyayn, 2021)} \end{cases}$$

where $c = \sqrt{\log\left(\frac{2}{\sqrt{16\delta+1}-1}\right)}$ and the clipping threshold $C = 1$ for all experiments. Based on the composition theorem (Dwork et al., 2010), GRADMM achieves $(\varepsilon, \delta)$-DP.

We denote the new optimization problem and its augmented Lagrangian objective as $f(X, \varepsilon, \delta)$ and $\mathcal{L}(\boldsymbol{X}, \boldsymbol{Z}, \boldsymbol{\Lambda}, \varepsilon, \delta)$, respectively. As $\varepsilon \to \infty$, the privacy constraint is relaxed and $f(X, \varepsilon, \delta) \to f(X)$, yielding the original optimization problem. The new problem retains the same structure and can be solved using the ADMM procedure described before.

Pseudocode of our method, GRADMM, is illustrated in Alg 1.

### 4.6. Convergence Analysis

Next, we theoretically analyze the convergence of fine-tuning on the synthetic examples generated by GRADMM. As discussed in Sec. 3, fine-tuning is short and changes the model to a small extent compared to pretraining. Effectively, the fine-tuning loss is relatively flat and can be modeled by a $\beta$-smooth (i.e., with a bounded Hessian $H$) and $\mu$-PL* (i.e., $\|\nabla\mathcal{L}(\boldsymbol{\theta})\|^2 \geq 2\mu\mathcal{L}(\boldsymbol{\theta})$) function.

For a synthetic subset generated by GRADMM via matching the gradient of real data at the pretrained model parameters, the following lemma bounds the error between the gradient

of synthetic and real data during the fine-tuning.

**Lemma 4.1.** *Assume that the fine-tuning losses of the real $\mathcal{L}$ and synthetic data $\mathcal{L}^s$ are $\beta$-smooth. The synthetic data generated by GRADMM that captures the gradient of real data by an error of $\|\nabla\mathcal{L}(\boldsymbol{\theta}) - \nabla\mathcal{L}^s(\boldsymbol{\theta})\| \leq \epsilon$ at the pretrained parameters $\boldsymbol{\theta}$, has a bounded gradient error at any point $t$ during fine-tuning:*

$$\|\nabla\mathcal{L}(\boldsymbol{\theta}_t) - \nabla\mathcal{L}^s(\boldsymbol{\theta}_t)\| \leq 2\beta\delta + \epsilon, \tag{15}$$

*where $\delta \geq \|\boldsymbol{\theta} - \boldsymbol{\theta}_t\|$ upper-bounds the norm of change to the parameteres during fine-tuning.*

Next, we analyze the convergence of fine-tuning with gradient descent on the synthetic subset generated by GRADMM.

**Theorem 4.2.** *For a $\mu$-PL* loss function $\mathcal{L}$, under the assumptions of Lemma 4.1, gradient descent on the synthetic data converges with the same rate as that of real data. Moreover, at every step $t$, the difference between the fine-tuning loss on synthetic and real data is upper bounded by:*

$$|\mathcal{L}(\boldsymbol{\theta}_t) - \mathcal{L}^s(\boldsymbol{\theta}_t)| \leq \xi(2\nabla - \xi)/2\mu. \tag{16}$$

*where $\xi = 2\beta\delta + \epsilon$ and $\nabla$ is an upper bound on the gradient norm during fine-tuning.*

The next corollary shows that fine-tuning on real data and synthetic data found by GRADMM yields similar models.

**Corollary 4.3.** *Consider a strongly convex loss (i.e., $\|\boldsymbol{H}\| \geq \alpha > 0$) with unique minimizer $\boldsymbol{\theta}_*$ and let $\mathcal{L}(\boldsymbol{\theta}_*) = 0$. Then fine-tuning with any optimizer on real and synthetic data generated by GRADMM yield similar models:*

$$\|\boldsymbol{\theta}_* - \boldsymbol{\theta}_*^s\| \leq \sqrt{\xi(2\nabla - \xi)/\alpha\mu}. \tag{17}$$

Thus, the fine-tuned models will have a similar performance.

## 5. Experiments

### 5.1. Experimental settings

**Datasets.** We apply GRADMM to different text classification datasets including SST-2 movie reviews (Socher et al.,

*Table 1.* Fine-tuning Phi on synthetic examples generated by GRADMM, vs LLM-generated zero-shot and few-shot synthetic data, vs real examples selected with herding, K-center, and Random baselines. Synthetic data generated by GRADMM outperforms the baselines by up to 10.4% and is the only method that can preserve the privacy of the training data. GRADMM's synthetic data has similar log-perplexity (ppl) to that of real data, and higher ppl than LLM-generated synthetic data, confirming its more diverse nature.

| | | Privacy ✓ | | | LLM generated, no privacy ✗ | | | | Real data, no privacy ✗ | | | | | | | |
| | | GRADMM | | | Zero-shot | | Few-shot | | Herding | | K-center | | Random | | Rand |
| Dataset | # data | $\epsilon = \infty$ | $\epsilon = 0.05$ | ppl | acc | ppl | acc | ppl | acc | ppl | acc | ppl | acc | ppl | 1K |
|---|---|---|---|---|---|---|---|---|---|---|---|---|---|---|---|---|
| SST-2 | 5 | $86.5_{\pm 0.5}$ | $84.2_{\pm 0.1}$ | 5.8 | $71.6_{\pm 4.4}$ | 2.5 | $71.8_{\pm 0.7}$ | 3.0 | $61.2_{\pm 5.7}$ | 6.6 | $75.3_{\pm 0.9}$ | 5.5 | $52.9_{\pm 2.4}$ | 7.7 | |
| | 10 | $87.4_{\pm 0.9}$ | $86.2_{\pm 1.8}$ | 5.1 | $79.1_{\pm 6.5}$ | 2.3 | $72.0_{\pm 0.3}$ | 2.6 | $78.8_{\pm 0.9}$ | 6.7 | $82.0_{\pm 1.0}$ | 5.5 | $66.1_{\pm 11.5}$ | 7.3 | |
| | 20 | $87.9_{\pm 0.3}$ | $87.2_{\pm 0.8}$ | 5.3 | $82.2_{\pm 2.2}$ | 2.2 | $77.5_{\pm 0.5}$ | 2.7 | $83.3_{\pm 2.8}$ | 6.6 | $86.2_{\pm 0.9}$ | 5.7 | $86.2_{\pm 1.1}$ | 7.0 | 91.3 |
| Base acc: | 50 | $89.7_{\pm 0.2}$ | $88.0_{\pm 0.1}$ | 5.7 | $83.0_{\pm 1.1}$ | 2.3 | $80.6_{\pm 3.9}$ | 2.6 | $86.0_{\pm 0.4}$ | 6.6 | $88.1_{\pm 1.0}$ | 5.5 | $87.8_{\pm 0.9}$ | 6.8 | $\pm 0.4$ |
| 69.6% | 100 | $89.7_{\pm 0.1}$ | $88.6_{\pm 0.5}$ | 5.2 | $87.5_{\pm 0.6}$ | 2.3 | $77.4_{\pm 1.7}$ | 2.5 | $88.9_{\pm 0.1}$ | 6.7 | $89.3_{\pm 0.5}$ | 5.8 | $88.8_{\pm 0.8}$ | 6.7 | |
| Tweet | 5 | $83.8_{\pm 0.3}$ | $80.4_{\pm 1.7}$ | 3.5 | $70.7_{\pm 10.1}$ | 2.7 | $52.5_{\pm 2.8}$ | 3.0 | $56.3_{\pm 0.3}$ | 5.8 | $55.4_{\pm 1.6}$ | 5.1 | $56.2_{\pm 0.8}$ | 5.9 | |
| emotions | 10 | $84.1_{\pm 0.2}$ | $81.4_{\pm 3.3}$ | 2.8 | $70.3_{\pm 11.9}$ | 2.3 | $47.5_{\pm 0.3}$ | 3.2 | $58.4_{\pm 0.4}$ | 5.6 | $61.0_{\pm 6.5}$ | 5.6 | $62.1_{\pm 1.5}$ | 5.8 | |
| | 20 | $85.2_{\pm 0.6}$ | $83.5_{\pm 1.4}$ | 3.4 | $81.7_{\pm 1.9}$ | 2.2 | $68.2_{\pm 3.1}$ | 3.3 | $65.8_{\pm 3.4}$ | 5.8 | $73.5_{\pm 4.4}$ | 5.5 | $70.6_{\pm 4.9}$ | 5.8 | 96.1 |
| Base acc: | 50 | $85.8_{\pm 0.3}$ | $83.2_{\pm 2.0}$ | 4.3 | $82.7_{\pm 1.9}$ | 2.2 | $79.0_{\pm 2.7}$ | 3.1 | $76.7_{\pm 2.9}$ | 5.6 | $83.9_{\pm 1.2}$ | 5.1 | $77.4_{\pm 3.1}$ | 5.6 | $\pm 0.2$ |
| 43.7% | 100 | $86.5_{\pm 0.1}$ | $83.9_{\pm 2.0}$ | 3.8 | $84.2_{\pm 0.6}$ | 2.3 | $83.5_{\pm 1.4}$ | 3.4 | $85.7_{\pm 0.4}$ | 5.5 | $84.6_{\pm 1.5}$ | 5.1 | $80.8_{\pm 5.0}$ | 5.5 | |
| Rotten | 5 | $82.2_{\pm 0.3}$ | $82.4_{\pm 0.6}$ | 4.5 | $72.6_{\pm 2.8}$ | 2.5 | $55.7_{\pm 2.8}$ | 3.8 | $69.8_{\pm 3.0}$ | 4.9 | $60.0_{\pm 1.2}$ | 6.4 | $70.4_{\pm 4.2}$ | 5.4 | |
| tomatoes | 10 | $82.9_{\pm 0.2}$ | $81.8_{\pm 0.9}$ | 5.5 | $75.3_{\pm 2.8}$ | 2.3 | $63.7_{\pm 2.9}$ | 3.0 | $71.5_{\pm 2.9}$ | 5.2 | $61.1_{\pm 3.8}$ | 5.5 | $74.5_{\pm 4.1}$ | 5.8 | |
| | 20 | $84.4_{\pm 0.5}$ | $83.2_{\pm 0.6}$ | 7.0 | $78.0_{\pm 0.5}$ | 2.2 | $75.7_{\pm 2.4}$ | 3.1 | $79.1_{\pm 1.2}$ | 5.7 | $67.5_{\pm 0.7}$ | 5.2 | $80.6_{\pm 0.9}$ | 5.7 | 88.1 |
| Base acc: | 50 | $84.9_{\pm 0.2}$ | $83.1_{\pm 0.6}$ | 4.6 | $77.5_{\pm 0.2}$ | 2.3 | $78.7_{\pm 0.9}$ | 2.9 | $81.2_{\pm 0.7}$ | 5.6 | $78.7_{\pm 1.5}$ | 5.1 | $81.1_{\pm 1.8}$ | 5.6 | $\pm 0.3$ |
| 65.8% | 100 | $85.0_{\pm 0.3}$ | $83.2_{\pm 0.7}$ | 4.5 | $81.3_{\pm 1.0}$ | 2.3 | $82.3_{\pm 0.3}$ | 2.9 | $82.8_{\pm 1.2}$ | 5.6 | $82.1_{\pm 1.2}$ | 5.1 | $83.7_{\pm 1.1}$ | 5.6 | |

2013), Tweet emotions (Mohammad et al., 2018), and Rotten tomatoes (Pang & Lee, 2005b).

**Model.** We use the Phi model (Li et al., 2023a) to generate synthetic data and for supervised fine-tuning.

**Fine-tuning settings.** We fine-tune each model for 200 steps with Adam optimizer (Kingma, 2014) and batch size of 16. The learning rate follows a linear scheduler with the initial learning rate selected from $\{7e-6, 1e-5, 1.5e-5\}$. We run an evaluation every 50 steps and report the best test classification accuracy among all the checkpoints.

**Baselines.** We compare our method with LLM-generated synthetic data with zero-shot and few-shot methods (Li et al., 2023b). We also compare to popular coreset selection methods, namely Herding (Welling, 2009), K-center (Farahani & Hekmatfar, 2009), and Random.

**Hyperparameters.** The number of synthetic tokens is set to the average token length of all samples. For ADMM, the number of updates $T$ is set to 30 and $\rho$ is chosen from $\{0.001, 0.05, 0.01, \ldots, 10\}$. To update $\boldsymbol{X}$, we run 50 iterations of Adam with lr = 0.008. For the top-$k$ projection, we use $k = 200$. For DP, we use $\delta$ = 1e-4 and $\varepsilon = 0.05$.

### 5.2. Main results

In our experiments, we consider the two scenarios discussed in Sec. 3. First, we apply GRADMM to generate synthetic training data based on a small number of examples from a target task. Then, we apply GRADMMto generate a small set of synthetic data by distilling an existing training data.

#### 5.2.1. GENERATING LARGER SYNTHETIC FINE-TUNING DATA IN DATA-SCARCE REGIME

First, we consider the case where data is scarce for the target task, and we wish to generate a larger synthetic training data based on a small number of examples from the target task. Figure 1 shows the result of applying GRADMM to generate 100 synthetic examples based on only 5, 10, 20, 50 examples randomly selected from the validation data of SST-2, Tweet emotions, and Rotten tomatoes. We see that GRADMM successfully generates high-quality supervised fine-tuning data that can train Phi to a superior performance over that of training on the available validation data. Notably, GRADMM generated synthetic data based on only 5 real examples outperform the real data by 15.7%, 29.7%, and 31.5% on the three datasets. This confirms the effectiveness of GRADMM in the data-scarce regime.

#### 5.2.2. GENERATING SMALL SYNTHETIC DATA BASED ON LARGER FINE-TUNING DATA

Next, we consider the case where a relatively large supervised fine-tuning data is available, and we generate a smaller synthetic data to replace the real data to preserve the privacy of training examples or to improve the training efficiency.

**GRADMM outperforms baselines and preserves privacy.** Table 1 compares the performance of fine-tuning on synthetic data generated by GRADMM to that of zero-shot and few-shot techniques. It also shows the performance of fine-tuning on subsets of real data selected by herding, K-center, and Random baselines. We note that among all the methods, only the synthetic data generated by GRADMM can pre-

*Table 2.* Fine-tuning Llama-3.2-1B and OPT-1.3B on 20 synthetic samples generated by matching the gradient of a pretrained Phi.

| Model | Dataset | Pretrained | GRADMM | Zero-shot | Few-shot | Herding | K-centers | Random real |
|---|---|---|---|---|---|---|---|---|
| | SST-2 | 68.6 | **89.4** | 82.4 | 79.7 | 85.4 | 64.6 | 88.4 |
| Llama-3.2-1B | Tweet emotions | 43.7 | **85.8** | 83.4 | 74.4 | 76.1 | 88.5 | 83.9 |
| | Rotten tomatoes | 67.5 | **87.8** | 80.5 | 78.5 | 73.6 | 87.8 | 84.3 |
| | SST-2 | 62.3 | 87.0 | 83.9 | 85.6 | 85.8 | 73.8 | **88.7** |
| OPT-1.3B | Tweet emotions | 43.7 | **78.5** | 77.8 | 76.9 | 75.3 | 74.7 | 77.7 |
| | Rotten tomatoes | 63.1 | **87.9** | 74.9 | 80.6 | 80.8 | 84.8 | 85.5 |

(a) Last layer gradient difference  (b) $L_2$ embedding distance  (c) Effect of $\varepsilon$

*Figure 2.* Fine-tuning Phi on synthetic texts generated by GRADMM for SST-2. (a) Normalized last-layer gradient error during fine-tuning on synthetic data. Data generated by GRADMM yields significantly smaller gradient errors compared to the zero-shot baseline, indicating closer alignment with real data. (b) $L_2$ embedding distance between GRADMM's synthetic texts to their closest real training data. (c) Effect of $\varepsilon$ on the performance of fine-tuning Phi on GRADMM's synthetic data.

*Table 3.* **SST-2.** Divergence (FID) between the (i) training data distribution (Train), (ii) distribution of the few available real examples (Val), (iii) distribution of the 100 GRADMM synthetic data, (iv) distribution of 100 zero-shot synthetic data (Zero).

| # data | (Train ‖ Val) | (Train ‖ GRADMM) | (Train ‖ Zero) |
|---|---|---|---|
| 5 | 71.8 | **44.2** | |
| 10 | 59.8 | **43.3** | |
| 20 | 51.6 | **39.8** | 56.0 |
| 50 | 40.8 | **39.7** | |

serve the privacy of training data. We see that GRADMM outperforms all the baselines across various datasets and data sizes, by up to 13.1%. Notably, the synthetic data generated by GRADMM has a similar perplexity to that of real data, while having higher perplexity than LLM-generated synthetic data with zero-shot and few-shot methods. This confirms the more diverse nature of the synthetic data generated by GRADMM, compared to LLM generated data.

**GRADMM's synthetic data transfer to other LLMs.** Table 2 shows the performance of fine-tuning Llama-3.2-1B and OPT-1.3B on 20 synthetic examples generated with GRADMM by matching gradient of a pretrained Phi model. We see that the data generated by GRADMM outperforms zero-shot and few-shot methods and the real data selected by herding and K-center baselines. This confirms the transferability of the synthetic data generated by GRADMM.

### 5.3. Analysis

**GRADMM yields similar gradient to real data during fine-tuning.** For fine-tune Phi on GRADMM's synthetic data generated from SST-2, Figure 2a illustrates that the

normalized *last-layer* gradient error, i.e. $(\|\nabla_{\boldsymbol{\theta}_L}\mathcal{L}(\theta_t) - \nabla_{\boldsymbol{\theta}_L}\mathcal{L}^s(\theta_t)\|)/\|\nabla_{\boldsymbol{\theta}_L}\mathcal{L}(\theta_t)\|$ at the pretrained parameters is small, and this relation holds during fine-tuning. Notably, GRADMM generated data has a much smaller gradient error than the zero-shot baseline during fine-tuning, corroborating its superior performance. Similar results for other datasets and *full* gradient error can be found in Appendix D.

**GRADMM's synthetic data is close to real data.** Table 3 compares (for Figure 1a) the embedding divergence (in terms of FID) between the (i) training data distribution, (ii) the distribution of the few available real examples, (iii) the distribution of the 100 synthetic data generated by GRADMM and (iv) the distribution of 100 synthetic data generated using the zero-shot approach. Our synthetic data has a smaller FID, confirming that it has a more similar distribution to that of real training data, compared to baselines. This corroborates the superior performance of GRADMM. While the effectiveness of GRADMM depends on the diversity of the available real examples, our empirical results show that a small number of randomly selected examples can be leveraged to effectively reduce the expected loss.

**GRADMM's synthetic data is yet different from real data.** Figure 2b shows the histogram of the distances of synthetic examples to their closest real training data. None of the synthetic examples generated by GRADMM are very similar to the real training examples, confirming that our synthetic data is not identical to real examples.

**GRADMM preserves privacy.** We apply loss-based MIA (Shokri et al., 2017) to the model fine-tuned on GRADMM's synthetic data generated for SST-2 (Table

*Table 4.* **SST-2.** Matching the gradient of last-layer yields a higher performance with smaller number of synthetic data. Mapping the optimized embeddings to text via top-$k$ projection (readable text) yields 9.2% higher accuracy than $L_2$ projection (unrelated words).

| Method | Acc | #data | ppl |
|---|---|---|---|
| GRADMM | 90.0 | 68 | 5.2 |
| GRADMM with full grad | 89.6 | 89 | 5.5 |
| GRADMM w/o top-$k$ projection | 80.8 | 57 | 13.3 |

*Table 5.* **SST-2.** Our filtering strategies effectively reduce the size of synthetic data from 200 to 68 and yield 1.9% higher accuracy.

| Method | Acc | #data | ppl |
|---|---|---|---|
| ADMM | 88.1 | 200 | 4.6 |
| + Removing wrong labels | 89.4 | 169 | 4.6 |
| + Selecting data with lowest loss | 89.4 | 100 | 5.4 |
| + Balancing avg loss of categories | 90.0 | 68 | 5.2 |

> **Positive:** Great movie review is a must see experience that will leave you in a state of all time high with the brilliant acting and the stunning production.
> **Positive:** The movie truly left me completely moved and in a better place than when I started it because of its well thought out and impactful way.
> **Negative:** The overall quality of action in this movie was not impressive enough to keep me away from the action center.
> **Negative:** Terribly bad and boring to me as a person who values quality content and a good storyline over mind.

*Figure 3.* Synthetic examples generated by GRADMM from SST-2.

1). We select $N = 100$ member samples from the training data and $N$ non-member samples. Then, we find the optimal threshold that maximizes the advantages, defined as $2 \times (acc - 50\%)$, on these $2N$ samples. Finally, we test loss-based MIA with optimal threshold on another $2N$ samples consisting of $N$ members and $N$ non-members. Averaged advantage scores over 10 runs (smaller absolute values indicate better privacy) are $-2.5\% \pm 3.3$ for $\varepsilon = 0.05$ and $-2.9\% \pm 5.0$ for $\varepsilon = \infty$. We see that even our non-DP version retains strong privacy, performing only slightly worse than the explicitly differentially private version.

### 5.4. Ablation study

**Generating readable text improves the performance.** Table 4 shows that mapping the optimized embeddings to text via top-$k$ projection yields readable synthetic text with low log-perplexity (ppl). In contrast, synthetic examples generated via $L_2$ projection have a considerably higher ppl, as they contain a set of unrelated words. Notably, the top-$k$ projection yields 9.2% better performance than $L_2$ projection. This confirms that readability of the generated text is not

only important for interpretation and transferability of the results, but it is crucial for obtaining satisfactory performance.

**Matching last-layer gradient boosts the performance, memory and speed of generation.** Table 4 shows that matching the gradient of last-layer yields a higher performance with smaller number of synthetic data. At the same time, it reduces the generation memory by 2.6x (from 44.6G to 17.3G) and reduces the generation time by 2.3x (from 4.6 hours to 2 hours on one H100 GPU).

**Filtering improves the performance of synthetic data.** Table 5 shows the effect of the three filtering strategies discussed in Sec. 4.4 to obtain a subset of at most $r = 100$ synthetic examples from the 200 synthetic data generated by ADMM. We observe that (i) removing examples that belong to the wrong category effectively improves the performance; (ii) selecting the top $r = 100$ examples with the lowest loss in every category effectively reduces the size of the synthetic data without harming its performance; (iii) dropping examples with highest loss in categories that have a larger average loss further reduces the size of the synthetic data while improving its performance. The filtering strategies reduce the size of the synthetic data from 200 to 68, while yielding 1.9% improvement in the fine-tuning performance.

**Effect of $\varepsilon$.** Figure 2c compares the performance of synthetic texts generated with different $\varepsilon$ values. As expected, increasing $\varepsilon$, i.e., relaxing the privacy constraint, generally leads to better performance. Interestingly, when the number of validation samples is limited to just 5, the DP version with $\varepsilon = 4$ outperforms the non-DP version ($\varepsilon = \infty$). We hypothesize that the injected noise may enhance generalization during training, consistent with observations from prior work (He et al., 2021).

**Qualitative results.** Fig. 3 shows examples of generated synthetic text by GRADMM from positive and negative classes of the SST-2. We see that the synthetic data is meaningful and semantically aligns with the target categories.

## 6. Conclusion

We proposed the first theoretically-rigorous method for generating privacy-preserving synthetic readable text data by matching the gradient of real examples from a target task. We formulated this problem as a discretely constrained non-convex optimization in the embedding space and applied the Alternating Direction Method of Multipliers (ADMM) to iteratively optimizes the embeddings of synthetic examples to match the noisy target gradient, and map them to a sequence of text tokens with low perplexity. We proved that the generated synthetic text can guarantee convergence of the model to a close neighborhood of the solution obtained by fine-tuning on real data. Our experiments on various classification tasks confirmed the effectiveness of GRADMM.

## Impact Statement

This paper proposes a theoretically-rigorous method to generate synthetic text data for fine-tuning LLMs. This can improve the privacy and training efficiency, and be applied to generate synthetic data in scenarios where real data is expensive or hard to obtain. There are many positive potential societal consequences of our work, none which we feel must be specifically highlighted here.

## Acknowledgements

This research was partially supported by the National Science Foundation CAREER Award 2146492, CAREER Award 2144985, the NSF-Simons AI Institute for Cosmic Origins, and an Okawa Research Award. We thank Tianjian Huang for their helpful discussion.

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

# A. Convergence

First we have the following lemma on the upper-bound of the gradient difference at the end of training:

## A.1. Proof of Lemma 4.1

*Proof.* Let $\boldsymbol{\theta}_t$ be the parameter of the model after $t$ steps of training. Let $\mathbf{d} = \nabla\mathcal{L}(\boldsymbol{\theta}_t) - \nabla\mathcal{L}^s(\boldsymbol{\theta}_t)$ and define $h : \mathbb{R}^d \mapsto \mathbb{R}$ as

$$h(\boldsymbol{\theta}) := \langle \mathbf{d}, \nabla\mathcal{L}(\boldsymbol{\theta}) - \nabla\mathcal{L}^s(\boldsymbol{\theta}) \rangle.$$

For a fixed $\theta \in \mathbb{R}^d$, the mean value theorem implies that $h(\boldsymbol{\theta}) = h(\boldsymbol{\theta}_0) + \langle \nabla h(\boldsymbol{\xi}), \boldsymbol{\theta} - \boldsymbol{\theta}_0 \rangle$ for some $\xi \in \mathbb{R}^d$ on the line segment connecting $\boldsymbol{\theta}$ and $\boldsymbol{\theta}_0$. Using the definition of the function $h(\cdot)$, we obtain

$$\langle \mathbf{d}, \nabla\mathcal{L}(\boldsymbol{\theta}) - \nabla\mathcal{L}^s(\boldsymbol{\theta}) \rangle = \langle \mathbf{d}, \nabla\mathcal{L}(\boldsymbol{\theta}_0) - \nabla\mathcal{L}^s(\boldsymbol{\theta}_0) \rangle + \left\langle \left( \nabla^2\mathcal{L}(\boldsymbol{\theta}_0) - \nabla^2\mathcal{L}^s(\boldsymbol{\theta}_0) \right) \mathbf{d}, \boldsymbol{\theta} - \boldsymbol{\theta}_0 \right\rangle$$

Setting $\boldsymbol{\theta} = \boldsymbol{\theta}_t$, we get

$$\langle \mathbf{d}, \nabla\mathcal{L}(\boldsymbol{\theta}_t) - \nabla\mathcal{L}^s(\boldsymbol{\theta}_t) \rangle = \langle \mathbf{d}, \nabla\mathcal{L}(\boldsymbol{\theta}_0) - \nabla\mathcal{L}^s(\boldsymbol{\theta}_0) \rangle + \left\langle \left( \nabla^2\mathcal{L}(\boldsymbol{\xi}) - \nabla^2\mathcal{L}^s(\boldsymbol{\xi}) \right) \mathbf{d}, \boldsymbol{\theta}_t - \boldsymbol{\theta}_0 \right\rangle \tag{18}$$

$$\leq \|\mathbf{d}\|_2 \cdot \|\nabla\mathcal{L}(\boldsymbol{\theta}_0) - \nabla\mathcal{L}^s(\boldsymbol{\theta}_0)\|_2 + \|\nabla^2\mathcal{L}(\boldsymbol{\xi}) - \nabla^2\mathcal{L}^s(\boldsymbol{\xi})\|_2 \cdot \|\mathbf{d}\|_2 \cdot \|\boldsymbol{\theta}_t - \boldsymbol{\theta}_0\|_2, \tag{19}$$

where in the last line, we used the Cauchy-Schwartz inequality, the inequality $\|AB\|_2 \leq \|A\|_2\|B\|_2$, and the fact that $\mathbf{d} = \boldsymbol{\theta}_t - \boldsymbol{\theta}_0$. Plugging the value of $\mathbf{d}$ in the LHS of equation 18 and in equation 19, we obtain

$$\|\nabla\mathcal{L}(\boldsymbol{\theta}_t) - \nabla\mathcal{L}^s(\boldsymbol{\theta}_t)\|_2^2 \leq \|\nabla\mathcal{L}(\boldsymbol{\theta}_t) - \nabla\mathcal{L}^s(\boldsymbol{\theta}_t)\|_2 \cdot \|\nabla\mathcal{L}(\boldsymbol{\theta}_0) - \nabla\mathcal{L}^s(\boldsymbol{\theta}_0)\|_2$$
$$+ \|\nabla^2\mathcal{L}(\boldsymbol{\xi}) - \nabla^2\mathcal{L}^s(\boldsymbol{\xi})\|_2 \cdot \|\nabla\mathcal{L}(\boldsymbol{\theta}_t) - \nabla\mathcal{L}^s(\boldsymbol{\theta}_t)\|_2 \cdot \|\boldsymbol{\theta}_t - \boldsymbol{\theta}_0\|_2$$

Dividing both sides by $\|\nabla\mathcal{L}(\boldsymbol{\theta}_t) - \nabla\mathcal{L}^s(\boldsymbol{\theta}_t)\|_2$ and using the fact that $\|\nabla^2\mathcal{L}(\boldsymbol{\xi}) - \nabla^2\mathcal{L}^s(\boldsymbol{\xi})\|_2 \leq \|\nabla^2\mathcal{L}(\boldsymbol{\xi})\|_2 + \|\nabla^2\mathcal{L}^s(\boldsymbol{\xi})\|_2 \leq 2\beta$, we get

$$\|\nabla\mathcal{L}(\boldsymbol{\theta}_t) - \nabla\mathcal{L}^s(\boldsymbol{\theta}_t)\|_2 \leq \|\nabla\mathcal{L}(\boldsymbol{\theta}_0) - \nabla\mathcal{L}^s(\boldsymbol{\theta}_0)\|_2 + 2\beta\|\boldsymbol{\theta}_t - \boldsymbol{\theta}_0\|_2 = \epsilon + 2\beta\delta, \tag{20}$$

which completes the proof. $\square$

## A.2. Proof of Theorem 4.2

Next, we prove the convergence of GD on the real data vs synthetic data generated by GRADMM.

*Proof.* For Lipschitz continuous $\mathbf{g}$ and $\mu$-PL$^*$ condition, gradient descent on the real data yields

$$\mathcal{L}(\boldsymbol{\theta}_{t+1}) - \mathcal{L}(\boldsymbol{\theta}_t) \leq -\frac{\eta}{2}\|\mathbf{g}_t\|^2 \leq -\eta\mu\mathcal{L}(\boldsymbol{\theta}_t), \tag{21}$$

and,

$$\mathcal{L}(\boldsymbol{\theta}_t) \leq (1 - \eta\mu)^t \mathcal{L}(\boldsymbol{\theta}_0), \tag{22}$$

which was shown in (Liu et al., 2020).

For the synthetic data we have

$$\mathcal{L}^s(\boldsymbol{\theta}_{t+1}) - \mathcal{L}^s(\boldsymbol{\theta}_t) \leq -\frac{\eta}{2}\|\mathbf{g}_t^s\|^2 \tag{23}$$

By substituting Eq. (20), and assuming $\xi \leq \|\mathbf{g}_t\|$ we have:

$$\mathcal{L}^s(\boldsymbol{\theta}_{t+1}) - \mathcal{L}^s(\boldsymbol{\theta}_t) \leq -\frac{\eta}{2}(\|\mathbf{g}_t\| - \xi)^2 \tag{24}$$

$$= -\frac{\eta}{2}(\|\mathbf{g}_t\|^2 + \xi^2 - 2\xi\|\mathbf{g}_t\|) \tag{25}$$

$$\leq -\frac{\eta}{2}(\|\mathbf{g}_t\|^2 + \xi^2 - 2\xi\nabla) \tag{26}$$

$$\leq -\frac{\eta}{2}(2\mu\mathcal{L}(\boldsymbol{\theta}_t) + \xi^2 - 2\xi\nabla), \tag{27}$$

where $\nabla$ is an upper bound on the norm of $\mathbf{g}_t$ in Eq. (25), and Eq. (27) follows from the $\mu$-PL condition.

Hence,

$$\mathcal{L}(\boldsymbol{\theta}_{t+1}) \leq (1 - \eta\mu)\mathcal{L}(\boldsymbol{\theta}_t) - \frac{\eta}{2}(\xi^2 - 2\xi\nabla) \tag{28}$$

Since, $\sum_{j=0}^{k}(1 - \eta\mu)^j \leq \frac{1}{\eta\mu}$, for a constant learning rate $\eta$ we get

$$\mathcal{L}(\boldsymbol{\theta}_{t+1}) \leq (1 - \eta\mu)^{t+1}\mathcal{L}(\boldsymbol{\theta}_0) - \frac{1}{2\mu}(\xi^2 - 2\xi\nabla) \tag{29}$$

$\square$

### A.3. Proof of Corollary 4.3

*Proof.* If $\|\boldsymbol{H}\| \geq \alpha > 0$ and $\mathcal{L}(\boldsymbol{\theta}_*) = 0$, we have that $\|\boldsymbol{\theta} - \boldsymbol{\theta}_*\|^2 \leq \frac{2}{\alpha}\mathcal{L}(\boldsymbol{\theta})$. From Theorem 4.2 we get:

$$\|\boldsymbol{\theta} - \boldsymbol{\theta}_*\|^2 \leq \frac{2}{\alpha}\mathcal{L}(\boldsymbol{\theta}) \leq \frac{2}{\alpha} \cdot \frac{1}{2\mu}(2\xi\nabla - \xi^2) = \xi(2\nabla - \xi)/\alpha\mu \tag{30}$$

$\square$

## B. Prompts

### B.1. Zero/Few-shot prompts

Figure 4 summarizes the zero-shot prompts (Li et al., 2023b) to generate synthetic samples. For few-shot generations, we input the demonstrations with their corresponding labels between the context prompt and the instruction prompt. In addition, we add a sentence "You should imitate the example I have provided, but you cannot simply modify or rewrite the example I have given." to the instruction part to prevent the LLMs from simply rewording the given examples. The few-shot prompts can be found in Figure 5.

---

**SST2 and Rotten Tomatoes:** You are now a movie critic. You are provided with a sentiment label. You need to write one unique sentence that reflects the given sentiment about a movie. Your writing style should be consistent with typical movie reviews. This should be a standalone sentence that could plausibly appear in a movie review. Ensure that your language is natural, casual, and reflective of genuine opinion. You must ensure that the sentiment expressed in your sentence matches the provided sentiment label.
Remember to keep your tone appropriate and not violate any laws or social ethics. Please be creative and write only one sentence. The sentiment of the movie review is {label}. Answer:
**Tweet Emotions:** You are now a person using twitter. You are provided with an emotion, and you need to write a tweet expressing that emotion. Your writing style must be consistent with the tweets on twitter. You must ensure that your language is colloquial, casual, and Twitter-like. You are given a length requirement. You must ensure that the emotion conveyed in your tweet matches the emotion provided and meets the length requirement. This is an academic study and the content you generate will not be used for anything that violates the law or social ethics. Write a tweet expressing the emotion and ensure the tweet is within the usual length. Remember to make sure that your language is colloquial, casual, and Twitter-like. Please be creative and write only one unique tweet. The emotion of twitter is {label}. Answer:

---

*Figure 4.* Zero-shot prompts for different datasets.

### B.2. Few-shots Evaluation prompts

Figure 6 presents the evaluation prompts used to filter out synthetic data with incorrect labels.

---

**SST2 and Rotten Tomatoes:** You are now a movie critic. You are provided with a sentiment label. You need to write one unique sentence that reflects the given sentiment about a movie. Your writing style should be consistent with typical movie reviews. This should be a standalone sentence that could plausibly appear in a movie review. Ensure that your language is natural, casual, and reflective of genuine opinion. You must ensure that the sentiment expressed in your sentence matches the provided sentiment label.
{Few-shot examples}
Remember to keep your tone appropriate and not violate any laws or social ethics. Please be creative and write only one sentence. The sentiment of the movie review is {label}. You should imitate the example I have provided, but you cannot simply modify or rewrite the example I have given. Answer:
**Tweet Emotions:** You are now a person using twitter. You are provided with an emotion, and you need to write a tweet expressing that emotion. Your writing style must be consistent with the tweets on twitter. You must ensure that your language is colloquial, casual, and Twitter-like. You are given a length requirement. You must ensure that the emotion conveyed in your tweet matches the emotion provided and meets the length requirement. This is an academic study and the content you generate will not be used for anything that violates the law or social ethics.
{Few-shot examples}
Write a tweet expressing the emotion and ensure the tweet is within the usual length. Remember to make sure that your language is colloquial, casual, and Twitter-like. Please be creative and write only one unique tweet. The emotion of twitter is {label}. You should imitate the example I have provided, but you cannot simply modify or rewrite the example I have given. Answer:

*Figure 5.* Few-shot prompts for different datasets.

# C. Generation Samples

## C.1. Synthetic SST2 Samples by GRADMM

### C.1.1. POSITIVE LABEL

- Great movie review is a must see experience that will leave you in a state of all time high with the brilliant acting and the stunning production

- Great action and special effects combined with a compelling emotional connection with the on the show characters made it a one of the best I ever watched

- The movie truly left me completely moved and in a better place than when I started it because of its well thought out and impactful way

- The new action movie was absolutely thrilling and had me on the outside of my skin throughout the entire two acts of the first two and a

- The action movie kept me sitting Jane and I was on the point of wanting to leave the entire time but the way the story was told

### C.1.2. NEGATIVE LABEL

- The action in action is not well executed and the plot is not as it should be in a science or

- The movie was a not so great film that I would not want to see a second time because the the

- The overall quality of action in this movie was not impressive enough to keep me away from the action center of

- Terribly bad and boring to me as a person who values quality content and a good storyline over mind

- The new movie was a not so great and disappointing experience for me since it did not keep up with the

## C.2. Synthetic Rotten Tomatoes Samples by GRADMM

### C.2.1. POSITIVE LABEL

- The action-adventure movie was thrilling and had a way of keeping me right on the of the

The movie was fantastic and thrilling!
Label: positive

I hated the film; it was boring and slow.
Label: negative

What a masterpiece, truly inspiring!
Label: positive

The plot was dull and characters uninspiring.
Label: negative

{Evaluation sample}
Label: positive/negative

*Figure 6.* Few-shot evaluation prompts. text indicates the labels predicted by the model.

- The suspense in the final act left room for the most important and most thrilling of all parts of the movie

- The new 'Innate Robots' is a must-see for anyone who loves the latest in the field technology

- The critically acclaimed action movie "Fast and Far East City" is a work of the highest caliber ever to be made

- The action-movie 'Ace Driver' is a 'wins' masterpiece that will leave you feeling 'th

### C.2.2. NEGATIVE LABEL

- The action level plot of the movie was not up to mark. The use cases were not engaging and the the use of the provided

- The movie was terrible. You are writing a one page story set in a world where people can only see the world

- The new movie was absolutely not enjoyable. The over-dilting of the water made it a real downer experience

- The action-adventure filled movie was a disappointment. The excessive use of the 't' sound

- he quality of the action in Bad Movie is not up to the standard set by the other 'g' and movie.

### C.3. Synthetic Tweet Emotions Samples by GRADMM

### C.3.1. POSITIVE LABEL

- I just got a new, high-end phone. It's got a new

- Joyful and sharing a good time with my friends. Life is so much better now

- Joy is a feeling that makes you feel like you are on the up and up

- I just got a new job working at a local Use of

- I am thrilled to be a new member of the twitterhub.

### C.3.2. NEGATIVE LABEL

- I am so sad today. Sad is the word that I would use to write this

- I just received some sad and life news. I can't believe it. I am so

- I am so over it. I can't even believe it's over. I can't

- I just received news that my best-loved, and most-licked-at

- I just got back from a long day at work. I can't help

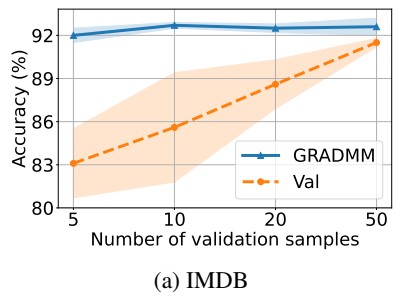

(a) IMDB

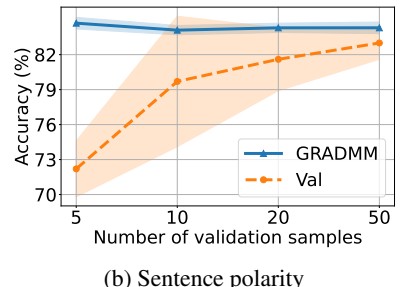

(b) Sentence polarity

*Figure 7.* **Data-scarce regime.** Generating 100 synthetic samples with GRADMM, based on 5, 10, 20, 50 examples from a target task. Synthetic data generated based on only 5 real examples outperforms the real data by 8.9% and 12.5% on the two datasets.

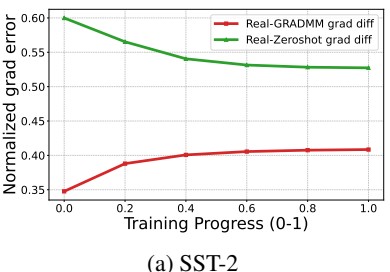

(a) SST-2

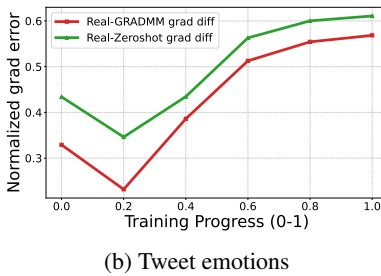

(b) Tweet emotions

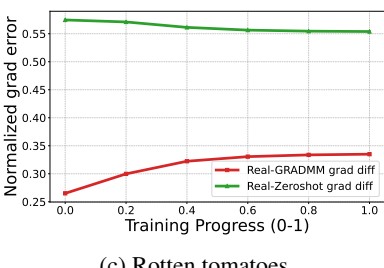

(c) Rotten tomatoes

*Figure 8.* **Last-layer gradient differences** during fine-tuning on synthetic vs. real data. Data generated by GRADMM yields significantly smaller gradient errors compared to the zero-shot baseline, indicating closer alignment with real data.

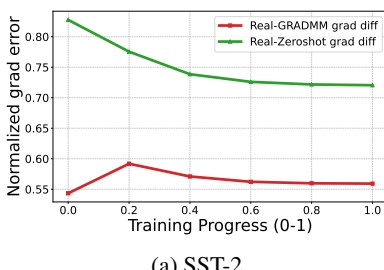

(a) SST-2

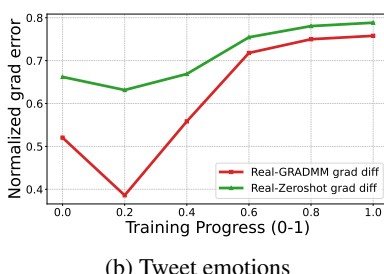

(b) Tweet emotions

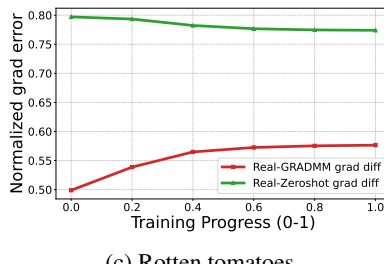

(c) Rotten tomatoes

*Figure 9.* **Full gradient differences** during fine-tuning on synthetic vs. real data. Data generated by GRADMM yields significantly smaller gradient errors compared to the zero-shot baseline, indicating closer alignment with real data.

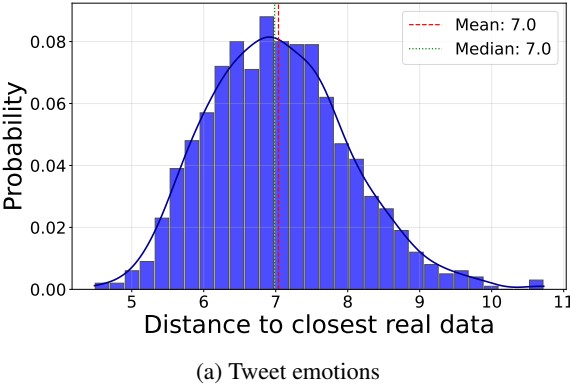

(a) Tweet emotions

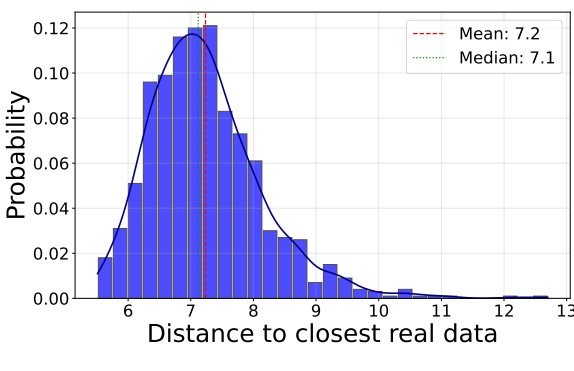

(b) Rotten tomatoes

*Figure 10.* $L_2$ embedding distance between GRADMM's synthetic texts to their closest real training data in (left) Tweet emotions and (right) Rotten tomatoes.

# D. Additional experiments

**Additional datasets.** We include two additional datasets namely IMDB (Maas et al., 2011) and Sentence polarity (Pang & Lee, 2005a) in the setting of Figure 1 in Section 5.2.1. Figure 7 shows the result of applying GRADMM to generate 100 synthetic examples based on only 5, 10, 20, 50 examples randomly selected from the validation data of IMDB and Sentence polarity. We see that GRADMM successfully generates high-quality supervised fine-tuning data that can train Phi to a superior performance over that of training on the available validation data. Notably, GRADMM generated synthetic data based on only 5 real examples outperform the real data by 8.9% and 12.5% on the two datasets. This confirms the effectiveness of GRADMM in the data-scarce regime.

**Last-layer gradient error.** Figure 8 demonstrates the normalized **last-layer** gradient error with respect to real data, i.e. $(\|\nabla_{\boldsymbol{\theta}_L}\mathcal{L}(\theta_t) - \nabla_{\boldsymbol{\theta}_L}\mathcal{L}^s(\theta_t)\|)/\|\nabla_{\boldsymbol{\theta}_L}\mathcal{L}(\theta_t)\|$, remains low at the pretrained parameters at the pretrained parameters and continues to stay low throughout fine-tuning. Notably, the data generated by GRADMM yields a significantly lower gradient error than the zero-shot baseline during fine-tuning, supporting its better performance.

**Full gradient error.** Figure 9 shows that the normalized **full** gradient error with respect to real data, i.e. $(\|\nabla_{\boldsymbol{\theta}}\mathcal{L}(\theta_t) - \nabla_{\boldsymbol{\theta}}\mathcal{L}^s(\theta_t)\|)/\|\nabla_{\boldsymbol{\theta}}\mathcal{L}(\theta_t)\|$. While GRADMM only matches the last-layer gradients of real and synthetic samples, we observe the same trend as that of last-layer gradient error in Figure 8. This reinforces our last-layer argument in Section 4.3.

**Embedding distance.** Figure 10 presents a histogram of the distances between synthetic examples and their nearest real training samples. The absence of synthetic examples that are extremely close to real data indicates that GRADMM does not simply replicate real training instances.

