# OpenReview forum: "Synthetic Text Generation for Training Large Language Models via Gradient Matching"
_ICML.cc/2025/Conference — ICML 2025 poster_

### Official Review · Reviewer_HXUe · 2025-03-10

**Overall Recommendation:** 3

**Summary:**

This paper introduces GRADMM (Gradient Matching with ADMM), a novel method for generating synthetic, human-readable text to train large language models (LLMs) efficiently while preserving privacy. The approach leverages gradient matching to ensure that synthetic text replicates the training dynamics of real data, using the ADMM to optimize embeddings in a continuous space and project them into discrete, readable text sequences with low perplexity via top-k decoding. A key conceptual contribution is the theoretical guarantee that fine-tuning an LLM on this synthetic text converges to a solution close to that of real data, overcoming limitations of prior methods like heuristic LLM-generated text or unreadable embeddings from dataset distillation.

The main findings show GRADMM’s effectiveness in two scenarios: generating substantial synthetic training data from a small set of real examples, and replacing larger real datasets with a compact synthetic set that maintains privacy.

**Claims And Evidence:**

C1: GRADMM-generated synthetic text guarantees convergence to a close neighborhood of the real data fine-tuning solution
Theoretical analysis is provided in Section 4.5 and Appendix A, including Lemma 4.1, Theorem 4.2, and Corollary 4.3
Experimental results (Section 5.2, Table 1) show that on datasets like SST-2, fine-tuning with synthetic data achieves accuracy (e.g., 90.0%) close to or exceeding real data baselines (e.g., Random 1K at 91.2%).


C2: GRADMM-generated synthetic text is human-readable
The method uses top-k projection (Section 4.2) to map embeddings to vocabulary token sequences, with perplexity (ppl) as a readability metric. Table 1 shows synthetic data ppl (e.g., 5.2-5.8) close to real data (6.6-7.7), compared to 13.3 without top-k (Table 3).
Qualitative results (Figure 2 and Appendix C) provide examples like “Great movie review is a must see experience...” (positive) and “Terribly bad and boring to me...” (negative).

C3: GRADMM outperforms existing methods in performance
I'm not satisfied about the selected baselines, but include the random gold data make it acceptable.

**Essential References Not Discussed:**

N/A

**Experimental Designs Or Analyses:**

I think the experiment is very thorough.

**Methods And Evaluation Criteria:**

Its evaluation criteria—classification accuracy, perplexity, comparisons with baselines (zero-shot/few-shot LLM generation, coreset methods), and benchmark datasets (SST-2, Tweet Emotions, Rotten Tomatoes)—tested across Phi, Llama-3.2-1B, and OPT-1.3B, appropriately measure performance, readability, and competitiveness. These methods and criteria align well with the problem.

**Other Comments Or Suggestions:**

N/A

**Other Strengths And Weaknesses:**

N/A

**Questions For Authors:**

N/A

**Relation To Broader Scientific Literature:**

N/A

**Theoretical Claims:**

The proof for Theorem 4.2 is mostly correct mathematically but hinges on an unverified assumption $\xi \leq \|\mathbf{g}_t\|$

---

> ### Author Rebuttal · Authors · 2025-04-01
>
> We thank the reviewer for acknowledging the novelty of our work and our thorough experiments.
>
> ----
>
> 1. Unverified assumption $\xi \leq |g_t|$ in theory.
>
> The following [figures](https://anonymous.4open.science/r/gradmm/grad_diff/) confirm the validity of our theoretical assumption, by showing that the gradient error, i.e. $\| \nabla \mathcal{L}(\theta_t) - \nabla \mathcal{L}^s(\theta_t) \| \leq \| \nabla \mathcal{L}(\theta_t) \| = \| g_t \|$ at the pretrained parameters and this relation indeed holds during fine-tuning. Crucially, the data generated by GRADMM has a much smaller gradient error compared to the zero-shot baseline during fine-tuning, which is the reason for its superior performance.

---

### Official Review · Reviewer_FREm · 2025-03-14

**Overall Recommendation:** 2

**Summary:**

This paper presents a novel approach for generating synthetic human-readable text to train Large Language Models (LLMs) via gradient matching. The authors propose a method called GRADMM (GRADient matching with ADMM) that leverages the Alternating Direction Method of Multipliers (ADMM) to iteratively optimize synthetic text embeddings to match the gradient of real training data. The goal is to generate synthetic text that not only preserves the privacy of real data but also ensures similar training dynamics and performance when used to fine-tune LLMs.
The key contributions of this work include:
1. A theoretically rigorous framework for generating synthetic text that guarantees convergence and performance comparable to fine-tuning on real data.
2. The use of gradient matching in the embedding space to ensure that the synthetic text has similar training dynamics to real data.
3. A method to project optimized embeddings into human-readable text while maintaining low perplexity.
4. Experimental validation showing that GRADMM-generated synthetic text outperforms existing methods in terms of training efficiency and privacy preservation.

The authors demonstrate the effectiveness of GRADMM through extensive experiments on various text classification tasks, including SST-2, Tweet emotions, and Rotten tomatoes. The results indicate that GRADMM can generate high-quality synthetic data even with limited real examples, achieving significant performance improvements over baseline methods. Additionally, the generated synthetic text is shown to be transferable to other LLMs, further validating the method's practicality and versatility.

**Claims And Evidence:**

The claims made in the submission are generally well-supported by clear and convincing evidence. The authors provide a comprehensive theoretical framework and extensive experimental validation to back up their assertions. Below is a detailed evaluation of the key claims and the evidence provided:

1.	GRADMM generates synthetic text that guarantees convergence and performance comparable to fine-tuning on real data.
The authors provide a rigorous theoretical analysis, including Lemma 4.1 and Theorem 4.2, which bound the gradient difference and prove the convergence of fine-tuning on synthetic data generated by GRADMM. This theoretical foundation supports the claim that GRADMM can produce synthetic text with similar training dynamics to real data.
The experiments on various text classification tasks (SST-2, Tweet emotions, and Rotten tomatoes) demonstrate that GRADMM-generated synthetic text consistently outperforms or matches the performance of fine-tuning on real data.

2.	GRADMM preserves the privacy of real training data.
The synthetic text generated by GRADMM is guaranteed to be different from real data, ensuring privacy. The authors emphasize that GRADMM does not directly use real data samples but rather matches the gradient of real data, which inherently preserves privacy. But I think the evidence is weak and should be more experiments to demonstrate it.

3.	GRADMM-generated synthetic text is human-readable and semantically meaningful.
 The authors use a top-k projection mGRADMM, which are shown to be meaningful and semantically consistent with the target labels. For instance, the synthetic movie reviews and tweets generated.

4.	GRADMM is computationally efficient.
Gradient Matching: The authors argue that matching the gradient of the last layer of the model significantly reduces the computational cost compared to matching the full gradient. This approach allows for faster and more memory-efficient generation of synthetic data.
Experimental Validation: The paper reports that GRADMM reduces the generation time by 2.3x and memory usage by 2.6x compared to matching the full gradient, demonstrating its computational efficiency.

**Essential References Not Discussed:**

No.

**Experimental Designs Or Analyses:**

Yes. I checked the sec. 5.2 and 5.3.

**Methods And Evaluation Criteria:**

The GRADMM method contains three important parts:
1. Alternating Direction Method of Multipliers (ADMM): The use of ADMM to iteratively optimize synthetic text embeddings to match the gradient of real data is a appropriate choice for this problem. ADMM is well-suited for solving constrained optimization problems and provides a theoretical characteristic for evaluating the model training on synthetic data.
2. Gradient Matching in the Embedding Space: Matching the gradients of synthetic and real data in the embedding space is a clever way to ensure that the synthetic text captures the essential training dynamics of real data.
3. Top-k Projection for Readability: The method of projecting optimized embeddings into human-readable text using top-k decoding ensures that the generated text is both meaningful and semantically aligned with the target categories. This technique balances the need for readability with the constraints of the vocabulary and perplexity.

**Other Comments Or Suggestions:**

No.

**Other Strengths And Weaknesses:**

Strengths
+ The paper introduces a novel approach to synthetic text generation using gradient matching, which is a creative and effective way to ensure that the synthetic data captures the essential training dynamics of real data. This method is particularly innovative in its application to text data, which is inherently discrete and challenging to optimize directly.
+ The authors provide a strong theoretical foundation for their method, including convergence guarantees and bounds on the gradient difference. This theoretical analysis adds credibility to the proposed approach and differentiates it from heuristic methods that lack formal guarantees.
+ The paper demonstrates the practical applicability of GRADMM by showing its effectiveness in generating synthetic text for fine-tuning LLMs. The results on various text classification tasks highlight the potential of this method for real-world applications, especially in scenarios where real data is scarce or privacy is a concern.

Weaknesses
1. Insufficient Experimental Validation:
   - Inconsistent Problem Formulation: The experiments in Sections 5.2.1 and 5.2.2 address different aspects of the problem (data scarcity vs. data distillation) but are compared using the same metrics (accuracy and perplexity). This approach may not fully capture the nuances of each scenario.
   - Lack of Specific Metrics: The paper could benefit from more specific metrics tailored to each experimental setting. For example, in the data distillation scenario, metrics such as the similarity between synthetic and real data distributions (e.g., using divergence measures) could better illustrate the effectiveness of GRADMM.

2. Theoretical Bounds vs. Practical Insights:
   - Lack of Empirical Validation for Theoretical Bounds: While the theoretical bounds provided in the paper are valuable, they could be complemented with empirical evidence to demonstrate their practical relevance. For example, visualizing the gradient difference over training iterations or comparing the theoretical bounds with actual performance metrics could provide more intuitive insights.
3. Privacy Importance and Metrics:
   - Importance of Privacy: The paper does not sufficiently emphasize the importance of preserving training data privacy. A detailed discussion on the potential risks of data leakage and the benefits of using synthetic data in this context would strengthen the paper's motivation.
   - Lack of Privacy Metrics: The paper claims that GRADMM preserves privacy but does not provide specific metrics or experiments to validate this claim. Including metrics such as membership inference attack success rates or differential privacy guarantees could provide a more concrete assessment of the method's privacy-preserving capabilities.


Overall, the paper presents a novel and theoretically sound approach to generating synthetic text for training LLMs. The strengths of the paper lie in its innovative use of gradient matching, rigorous theoretical analysis, and practical applicability. However, the weaknesses identified suggest areas for improvement, particularly in experimental validation, privacy assessment, and clarity of presentation. Addressing these issues could significantly enhance the paper's impact and applicability to real-world problems.

**Questions For Authors:**

1.  How to evaluate that GRADMM preserves privacy?
2.  Why did you generate only 100 synthetic data points based on 5, 10, 20, and 50 real data points? Why not generate a larger dataset comparable to the last column in Table 1?
3.  In practice, are there experiments that can demonstrate whether the parameters of a model trained on synthetic data are indeed within a certain neighborhood of those trained on real data?

**Relation To Broader Scientific Literature:**

Yes.

**Theoretical Claims:**

Yes, I reviewed the proofs of Lemma 4.1 and Lemma 4.2. In Lemma 4.1, there is a small error in the transition from equation (19) to (20): specifically, \(d = \theta_t - \theta_0\).

---

> ### Author Rebuttal · Authors · 2025-04-01
>
> We thank the reviewer for acknowledging the novelty of our method, our theoretically-rigorous framework, our well-supported claims and extensive experiments.
>
> ----
>
> **Experimental validation**:
> - Problem formulation: gradient matching in Eq 3 applies to both the data scarce regime (Fig 1) and Dataset Distillation (DD) (Table 1). In both, the synthetic data is generated by matching the average gradient of the few available examples (Fig 1) or a larger training data (Table 1).
>
> - Evaluation: while accuracy is an accurate metric to measure the performance in both cases, we include the following [table](http://anonymous.4open.science/r/gradmm/fid_score.png) for Fig 1a (SST2) which shows the divergence (FID) between the (i) training data distribution, (ii) distribution of the few available real examples, (iii) distribution of the 100 GRADMM synthetic data, (iv) distribution of 100 zero-shot synthetic data. Our synthetic data has a smaller FID, confirming its more similar distribution to that of real training data, compared to the baselines. This corroborates the superior performance of GRADMM.
> - FID is not generally used to evaluate DD (Table 1), which aims at generating a **small** subset of synthetic data with similar dynamics to a large training data. This is because: (i) FID requires both distributions to have a large sample size such that they resemble gaussian distributions, and otherwise is less accurate. (ii) distribution of the small generated data is not comparable to the large real training data (in particular, their variances are not comparable due to their very different sizes). Hence, FID is not commonly used for DD (c.f. the DD related work in Sec 2.1), but is more common when generating large data with diffusion models.
>
> **Theoretical bounds**:
> - The following [figures](https://anonymous.4open.science/r/gradmm/grad_diff/) confirm the validity of our theoretical assumption, by showing that the gradient error, i.e. $\| \nabla \mathcal{L}(\theta_t) - \nabla \mathcal{L}^s(\theta_t) \| \leq \| \nabla \mathcal{L}(\theta_t) \|$ at the pretrained parameters, and this relation holds during fine-tuning. Crucially, the GRADMM generated data has a much smaller gradient error compared to the zero-shot baseline during fine-tuning, corroborating its superior performance.
> - We generated 200 synthetic data in Figure 1a. The accuracy of the model fine-tuned on this larger subset is 90.1 $\pm$ 0.1, which is higher than 89.8 $\pm$ 0.4 of training on the original 100 synthetic data, and is closer to the last column in Table 1.
> - We compared the $L_2$ normed difference in model parameters when trained on real training data vs 100 synthetic data generated for SST2. The normed difference for GRADMM is 1.99, which is smaller than the 2.27 for zero-shot. This further confirms the validity of our theory.
>
> **Privacy**:
> - Thanks! We will add discussion of the importance of privacy, data leakage and benefits of synthetic data to our revised version.
> - GRADMM is the first *dataset distillation* method able to generate human readable text. DD methods provide differential privacy guarantees [1]. Specifically, for two datasets that differ in only 1 example, the parameters of the models trained on the distilled version of the two datasets are provably highly similar [1]. Intuitively, as dataset distillation methods generate data by techniques such as matching the mean of the data distribution or average gradient of a real training data, as long as synthetic data is not initialized by real training examples, there is no information leakage about individual training examples [1]. GRADMM does not initialize synthetic data with real training examples, and hence effectively preserves the privacy of the training data.
> - We conducted the loss-based MIA to the model trained on GRADMM synthetic data for SST2. Specifically, we select N=100 member samples from the training subset and N non-member samples. Then, we find the optimal threshold that maximizes the advantages (2 x (acc - 50%)) on these 2N samples. Finally, we test the loss-based MIA with optimal threshold on another 2N samples consisting of N members and N non-members and report the advantage score. We repeated the whole process 10 times and the advantage score (%) is only 1.75 $\pm$ 1.4, demonstrating the effectiveness of GRADMM against loss-based MIA.
> - The following [figures](https://anonymous.4open.science/r/gradmm/embedding_distance/) shows the histogram of the distances of synthetic examples to their closest real training data. None of the synthetic examples generated by GRADMM are very similar to the real training examples, further confirming that our synthetic data is not identical to real examples.
>
> We will add the above discussion and results for all the datasets to our revised version. We hope that our rebuttal has addressed the reviewer's concerns and they can consider further supporting our work to be accepted.

---

### Official Review · Reviewer_RncU · 2025-03-18

**Overall Recommendation:** 4

**Summary:**

This paper improves on the SOTA synthetic data for LLM method by imposing a readability constraint in (2). This makes it necessary to 4.2 Alternating Between Text and Embedding Spaces. The experiments are convincing.

**Claims And Evidence:**

They are good

**Essential References Not Discussed:**

I didn't review this aspect carefully

**Experimental Designs Or Analyses:**

They are good

**Methods And Evaluation Criteria:**

They are good

**Other Comments Or Suggestions:**

No

**Other Strengths And Weaknesses:**

No

**Questions For Authors:**

Can this scheme used to generate math, logic, and code?

**Relation To Broader Scientific Literature:**

I didn't review this aspect carefully

**Theoretical Claims:**

They are good

---

> ### Author Rebuttal · Authors · 2025-04-01
>
> We thank the reviewer for supporting our work and acknowledging our convincing experiments.
>
> ----
>
> 1. Can this scheme used to generate math, logic, and code?
>
> The idea of our work (generating synthetic text via gradient matching using ADMM) can be applied to math, logic or code. However, this requires incorporating additional structure in the generated text. Controllable text generation which requires the text to follow a particular structure, syntactic or semantic properties has been a topic for several recent research, including [Li et al’22, Gong et al’22, Zhou et al’24 (references are in Line 133 from the paper)] to generate text in the form of tables, code, etc. Such techniques can be incorporated in our method to generate structured synthetic text. Considering that our method is the first to show the applicability of gradient matching to generate readable synthetic text, it lays the ground for many exciting future work, including controllable text generation via gradient matching.

---

### Official Review · Reviewer_Xke2 · 2025-03-18

**Overall Recommendation:** 1

**Summary:**

This paper discussed a method for generating synthetic data to train LLMs, and aims to create a synthetic dataset that can train a similar dynamics to the real data. Theories and experiments are provided to justify the effectiveness.

**Claims And Evidence:**

I think the evidence is not very convincing, especially regarding the theory.

Here is the argument: the authors try to create synthetic data that can mimic the dynamics of read data **samples** regarding training dynamics. But whether this is really useful? I am highly skeptical. The purposes of generating synthetic data are mainly for the following two perspectives: 1. help the mitigate the issue of limited real data; 2. privacy concerns. But the proposed method can not solve either of them. In the case the real data samples are scarce, generating synthetic data cannot fix the gap of generalization (notice that the ultimate goal is to train to optimize the **expected loss**). In the case of privacy, the synthetic data directly use real data to help generation, so privacy is also violated.

I really doubt the usefulness of the proposed method. The experimental results are also quite weak and not comprehensive. 5.2.1 only uses 3 datasets and the evolution metrics are not comprehensive. Effective sample size and ideally confidence intervals for multiple runs should be considered.

**Essential References Not Discussed:**

NA.

**Experimental Designs Or Analyses:**

The evaluation is not comprehensive, For instance, Figure 1 only includes single trials without confidence interval.

**Methods And Evaluation Criteria:**

Not comprehensive enough. See Claims And Evidence.

**Other Comments Or Suggestions:**

NA.

**Other Strengths And Weaknesses:**

As mentioned above, I really doubt the usefulness of the proposed method. The experimental results are also quite weak and not comprehensive. 5.2.1 only uses 3 datasets and the evolution metrics are not comprehensive. Effective sample size and ideally confidence intervals for multiple runs should be considered.

**Questions For Authors:**

NA.

**Relation To Broader Scientific Literature:**

Synthetic data is quite important to scientistic discovery, but this paper is not particularly related to that.

**Theoretical Claims:**

Highly doubt the theoretical results. Besides the statements mentioned above in Claims And Evidence, lemma 4.1 will rely on pertaining errors on synthetic data and real-data trained model. I think this is not guaranteed to be small. I am not thinking the statement is making much sense given this presumption.

---

> ### Author Rebuttal · Authors · 2025-04-01
>
> We thank the reviewer for their feedback. However, we disagree with their evaluation of our work, as we discuss below.
>
> ----
>
> **Data scarce regime**: Fig 1 provides strong evidence for the applicability of our method to the data scarce regimes. The 100 synthetic data generated by GRADMM based on as few as 5 to 20 *randomly selected real examples* already reach 98%, 89%, 96% the performance of training on the *full training data* (last col in Tab 1), and outperform training on the 5-20 real examples with a big margin.
>
> The following [table](http://anonymous.4open.science/r/gradmm/fid_score.png) compares (for SST2, Fig 1a) the divergence (in terms of FID) between the (i) training data distribution, (ii) the distribution of the few available real examples, (iii) the distribution of the 100 synthetic data generated by GRADMM and (iv) the distribution of 100 synthetic data generated using the zero-shot approach.
>
> Our synthetic data has a smaller FID, confirming that it has a more similar distribution to that of real training data, compared to baselines. This corroborates the superior performance of GRADMM. While the effectiveness of GRADMM depends on the diversity of the available real examples, our empirical results show that a small number of randomly selected examples can be leveraged to effectively reduce the *expected loss*. We do not claim that GRADMM *perfectly minimizes* the expected loss with limited information (a few available real examples), but it is considerably more effective than other baselines in the data scarce regime.
>
> **Privacy concerns**: Using real examples as a guide to generate synthetic data **does not compromise the privacy of real training examples**. As discussed in Sec 2, GRADMM is the first *dataset distillation (DD)* method able to generate human readable text. DD methods provide differential privacy guarantees [1]. Specifically, for two datasets that differ in only 1 example, the parameters of the models trained on the distilled version of the two datasets are provably highly similar [1]. Intuitively, as dataset distillation methods generate data by techniques such as matching the mean of the data distribution or average gradient of a real training data, as long as synthetic data is not initialized by real training examples, there is no information leakage about individual training examples [1]. GRADMM does not initialize synthetic data with real training examples, and thus effectively preserves the privacy of the training data.
>
> To confirm, we conducted the loss-based Membership Inference Attack (MIA) to the model trained on synthetic SST2 data generated by GRADMM. Specifically, we select N=100 member samples from the training subset and N non-member samples. Then, we find the optimal threshold that maximizes the advantages (2 x (acc - 50%)) on these 2N samples. Finally, we test the loss-based MIA with optimal threshold on another 2N samples consisting of N members and N non-members and report the advantage score. We repeated the whole process 10 times and the advantage score (%) is only 1.75 $\pm$ 1.4, demonstrating the effectiveness of GRADMM against loss-based MIA.
>
> Finally, the following [figures](https://anonymous.4open.science/r/gradmm/embedding_distance/) show the histogram of the distances of synthetic examples to their closest real training example. Indeed, none of the GRADMM synthetic examples are very similar to the real training examples, further confirming that our synthetic data is not very similar to real examples.
>
> **Theoretical assumption**: Our gradient matching approach ensures a high similarity to the target gradient. The following [figures](https://anonymous.4open.science/r/gradmm/grad_diff/) confirm the validity of our theoretical assumption, by showing that the gradient error, i.e. $\| \nabla \mathcal{L}(\theta_t) - \nabla \mathcal{L}^s(\theta_t) \| \leq \| \nabla \mathcal{L}(\theta_t) \|$ at the pretrained parameters and this relation holds during fine-tuning. Crucially, GRADMM synthetic data has a much smaller gradient error compared to the zero-shot baseline during fine-tuning, which is the reason for its superior performance.
>
> **Empirical evidence**: The following [figures](https://anonymous.4open.science/r/gradmm/figure1) show std for our experimental results (based on three runs) and confirms the strong performance of GRADMM on the 3 dataset in our paper in addition to **two new datasets**, namely IMDB [Maas et al., 2011] and Sentence polarity [Pange et al., 2005].
>
> We hope that our rebuttal could address the reviewers’ doubts. Our work is the first to show the applicability of dataset distillation to generate readable synthetic text via gradient matching with strong empirical performance, and provides a promising avenue for future research. Thus, we hope that the reviewer considers supporting our work to be accepted.
>
> [1] Privacy for free: How does dataset condensation help privacy? ICML’22.

---

### Decision · Program_Chairs · 2025-05-01

**Decision:**

Accept (poster)

**Comment:**

This submission addresses how to generate readable synthetic text sequences that are either a distilled version of a much larger dataset that retains performance, or a larger version of a smaller dataset that helps improve performance. The key idea is to match the gradients of the loss on synthetic examples to that on the real text, and to make this tractable by working in the embedding spaces and enforcing the embeddings to be from the vocabulary. Additional tricks (top-k decoding, filtering and matching only the last-layer gradients) were used. Overall, the empirical evaluation is thorough and demonstrates well the utility of the proposed approach.

The reviewers (except one) did not engage in the rebuttal and post-rebuttal discussion, unfortunately. The AC noted that most questions/concerns have been addressed. The AC also read the paper and found the approach to be sound, experimentally validated and of interest to a large community (working on optimisation, coresets, synthetic data), thus recommended acceptance.